# Spatial and temporal dynamics of West Nile virus between Africa and Europe

Giulia Mencattelli [1,2,3,7] ✉, Marie Henriette Dior Ndione[4,7], Andrea Silverj[2,3,5,7], Moussa Moise Diagne[4], Valentina Curini[1], Liana Teodori[1], Marco Di Domenico [1], Rassoul Mbaye[4], Alessandra Leone[1], Maurilia Marcacci[1], Alioune Gaye[6], ElHadji Ndiaye[6], Diawo Diallo[6], Massimo Ancora[1], Barbara Secondini[1], Valeria Di Lollo[1], Iolanda Mangone[1], Andrea Bucciacchio[1], Andrea Polci[1], Giovanni Marini [3], Roberto Rosà [2,3], Nicola Segata [5], Gamou Fall[4], Cesare Cammà[1], Federica Monaco[1], Mawlouth Diallo[6], Omar Rota-Stabelli [2,3,5], Oumar Faye[4,8], Annapaola Rizzoli[3,8] & Giovanni Savini[1,8]

It is unclear whether West Nile virus (WNV) circulates between Africa and Europe, despite numerous studies supporting an African origin and high transmission in Europe. We integrated genomic data with geographic observations and phylogenetic and phylogeographic inferences to uncover the spatial and temporal viral dynamics of WNV between these two continents. We focused our analysis towards WNV lineages 1 (L1) and 2 (L2), the most spatially widespread and pathogenic WNV lineages. Our study shows a Northern-Western African origin of L1, with back-and-forth exchanges between West Africa and Southern-Western Europe; and a Southern African origin of L2, with one main introduction from South Africa to Europe, and no back introductions observed. We also noticed a potential overlap between L1 and L2 Eastern and Western phylogeography and two Afro-Palearctic bird migratory flyways. Future studies linking avian and mosquito species susceptibility, migratory connectivity patterns, and phylogeographic inference are suggested to elucidate the dynamics of emerging viruses.

West Nile virus (WNV) is one of the most widespread viruses in the world, whose geographic expansion has been associated with changes in ecological suitability including climate change[1,2]. Considered one of the main One Health challenges, this arthropod-borne virus belonging to the genus *Flavivirus*, family *Flaviviridae*, and Japanese Encephalitis serocomplex, is maintained in nature through a transmission cycle involving wild birds and vector-competent mosquitoes[3]. Although in the majority of the susceptible bird species infection generally does not lead to severe clinical signs, in some species it might cause severe

neurological complications, including death[4–6]. Competent vector mosquitoes, mainly belonging to the *Culex* (*Cx.*) genus, can transmit the virus to humans and other animal species, with an increasing rate and impact on human and animal health worldwide[7]. Infected humans are mostly asymptomatic, but in 20% of cases the infection can lead to a febrile illness (West Nile fever (WNF)), and in 1% of cases, mainly in elderly or immunocompromised people, to severe and sometimes fatal neurological disease (West Nile virus neuroinvasive disease (WNND))[8]. The main objective of timely WNV surveillance at the EU

[1]Istituto Zooprofilattico Sperimentale dell'Abruzzo e del Molise, Teramo, Italy. [2]Centre Agriculture Food Environment, University of Trento, San Michele all'Adige, Italy. [3]Research and Innovation Centre, Fondazione Edmund Mach, San Michele all'Adige, Italy. [4]Virology Department, Institut Pasteur de Dakar, Dakar, Senegal. [5]Department CIBIO, University of Trento, Trento, Italy. [6]Medical Zoology Department, Institut Pasteur de Dakar, Dakar, Senegal. [7]These authors contributed equally: Giulia Mencattelli, Marie Henriette Dior Ndione, Andrea Silverj. [8]These authors jointly supervised this work: Oumar Faye, Annapaola Rizzoli, Giovanni Savini. ✉e-mail: giulia.mencattelli@gmail.com

level is to provide early warning to public health professionals of areas with human WNV infections and thereby preventing human-to-human transmission via contaminated donations. The EU Blood Safety Directive requires blood establishments to defer donors for 28 days after leaving an area where human cases have been detected unless an individual donor nucleic acid test is negative[9]. WNV can also be a serious economic issue and source of emotional distress if horses are infected. In this species, infections can cause severe neurological disorders with a high mortality rate (33%)[10].

Phylogenetic analyses and phylogeographic inference reconstructions integrating detailed geographic observations at high-resolution scales can provide valuable insights into the spatial dynamics of pathogens in a geographic and temporal context[11]. Starting from viral genome data and inferring continuous phylogeographic diffusion through space and time, Bayesian statistical approaches are now valuable for monitoring the viral spread between different geographic areas of the world and providing relevant information for epidemic prediction and preparedness[12]. This approach is particularly interesting for viruses such as WNV which are carried over long distances across the globe by migratory birds, playing a crucial role in pathogen ecology, circulation, and spread[1]. Indeed, decades of molecular phylogenetic studies have shown that WNV is characterised by a high genetic diversity. At least nine lineages have been characterised worldwide, four of which circulate in Africa [lineage 1 (L1), lineage 2 (L2), lineage 7 (L7—now classified as the Koutango virus), and putative lineage 8 (L8)][10,13]. In Europe, L1 and L2 strains are the most prevalent[14]. Phylogenetic studies of WNV suggest links between Africa and Europe. It appears that some European epidemics were the result of the introduction of WNV African strains. With regard to WNV L1, the first introduction probably occurred more than 25 years ago from Northern-Western African countries to Italy or France, where the strain was first detected in 1998 and 2000, respectively[10,15]. WNV L2, on the other hand, was first introduced into Hungary in 2004[14], but its origin is still uncertain[10,16]. Following these initial introductions and probably after few other introductory events, both lineages spread, established, and co-circulated in many European countries. In 2022, they were responsible for 965 human cases and 115 deaths (https://www.ecdc.europa.eu/en/west-nile-fever/surveillance-and-disease-data/disease-data-ecdc)[10]. Furthermore, an increased pathogenicity has been reported for L1[17].

Despite the existence of several studies focusing on WNV circulation[10,11,13,18,19], little is known about the genetic relationships between European and African WNV L1 and L2 strains. This study aims to elucidate the pattern of virus circulation of the two lineages between Africa and Europe, the two key continents for studying the ecology and evolution of this pathogen. Specifically, we aim to (i) provide a new dataset of genomes, including sequences from Italy and Senegal collected between 2006 and 2022; and (ii) describe the dispersal dynamics of the two lineages in Africa and Europe through time and space. This information will be useful to the scientific community and public health authorities for future research studies and epidemic intelligence activities.

## Results
### Genome sequence analysis
Illumina sequencing produced an average total number of 2,330,817 trimmed reads. The number of mapped reads (151 nucleotides [nt] in length) ranged from 238,259 to 1,035,983, with coverage depth ranging from 2,770.45× to 6,862.12×. Consensus sequences were characterised from 19 WNV L2 Italian samples, collected between 2021 and 2022, and 7 Senegalese WNV L1 and 3 WNV L2 samples, collected between 2006 and 2016, that were uploaded to BankIT NCBI (https://submit.ncbi.nlm.nih.gov/about/bankit/) the 20th of June 2022, the 27th of October 2022, and the 15th of November 2022, respectively.

### Phylogenetic and phylogeographic inference of WNV
Phylogenetic and phylogeographic patterns were reconstructed separately from WNV L1 and L2 genome sequences. We detected a strong correlation between the genetic distance and the date through a root-to-tip regression analysis, which was 0.90 for the L1 clock dataset and 0.71 for the L2 clock dataset (Supplementary Data 5 and Supplementary Figs. 5 and 6). These values change only slightly in the reduced versions of the datasets (Supplementary Data 5). This denotes a clear and robust clock signal in our data, which justifies the assumption of a molecular clock.

For WNV L1, our maximum likelihood phylogeny and molecular clock analyses revealed that most American, European, Middle Eastern, Asian, and African strains are clustered inside one major clade, Clade 1A (Supplementary Fig. 1). Supplementary Fig. 1 identifies seven additional clusters within Clade 1A.

Cluster 1 contains the first WNV L1 strain recovered in Africa (Egypt) in 1951 (Genbank AF260968), as well as strains circulating in Israel (1953), Russia (1963), Azerbaijan (1967-70), India (1968, 2015), and Portugal (1971).

Cluster 2 contains more recent isolates. In particular, three major sub-clusters are defined within this cluster: i) the Western-Mediterranean (W-Med), with strains (in chronological order of detection) from Morocco (1996 and 2003), Italy (1998 − 2022), Israel (2000), France (2000, 2004, and 2015), Portugal (2004), Spain (2007, 2008, 2010, and 2020), and Senegal (2012 − 2018); ii) the Eastern European, with strains from Romania (1996) and Russia (1999 − 2006); and iii) a group of strains from Kenya (1998 and 2010), Senegal (1990), Zambia (2019), United Arab Emirates (2015), and Cyprus (2016). Our maximum likelihood analysis identifies two further major groups of sequences (WMed-1 and WMed-2) within the W-Med subtype. These groups differ slightly from the early WMed WNV strains of Morocco (1996), France and Israel (2000), already proposed by[20], strains collected in Italy between 2011 and 2013 (Livenza, Ancona, Piave), and the more divergent strain of Senegal of 2012 (Genbank ON813216) (Fig. 1 and Supplementary Fig. 1).

Most of the strains collected since 2003 from Morocco (2003), Portugal (2004), France (2004), Italy (2008 - 2022), Spain (2010 and 2020), and Senegal (2012 - 2018), are included into the WMed-1 sub-cluster. The WMed-2 includes one isolate from Italy (1998) and three from Spain (2007 and 2008) (Fig. 1). The WMed-1 and 2 appear to be all rooted in the 2000 French strain (Genbank AY268132).

Cluster 3 (Supplementary Fig. 1) includes strains from Russia (1999), India (2011), and China (2011) while cluster 4 contains a large group of strains from Senegal (1990), Tunisia (1997), America (1999-2019), Israel (1998 and 2000), Hungary (2003), and Belgium (2014). Cluster 5 includes strains from Nigeria and Senegal in 1965 and 1979, respectively. Cluster 6 contains strains from the Central African Republic (CAR; 1967) and Turkey (2011). Cluster 7 includes strains from Senegal (1996) and Spain (2017).

Our analysis shows that clusters 2, 3, 4, 6, and 7 are rooted by the two ancient sequences from Nigeria and Senegal included in cluster 5 (Fig. 1 and Supplementary Fig. 1). It is also shown that all strains within cluster 2 are rooted by the 1989 Senegalese strain (Genbank OP846971), with bootstrap support = 100. Furthermore, within cluster 2, the WMed-1 subtype shows a close genetic similarity with a group of ten strains of Senegal (2012-18) and one strain of France of 2015, which are clustered together and appear to be closely related to a group of 2021-22 Italian sequences (bootstrap support = 100; Supplementary Fig. 1). Finally, the WNV strains collected in 2020 and 2022 from Campania region (Italy) (Genbank MW627239 and OP850023) are closely related to one strain of Senegal of 2012 (Genbank ON813215), which in turn are genetically similar to a group of strains of Italy (2008, 2009 and 2011) and Spain (2020).

Our phylogeographic inference indicates that all strains included in Clade 1A to have originated from the African continent

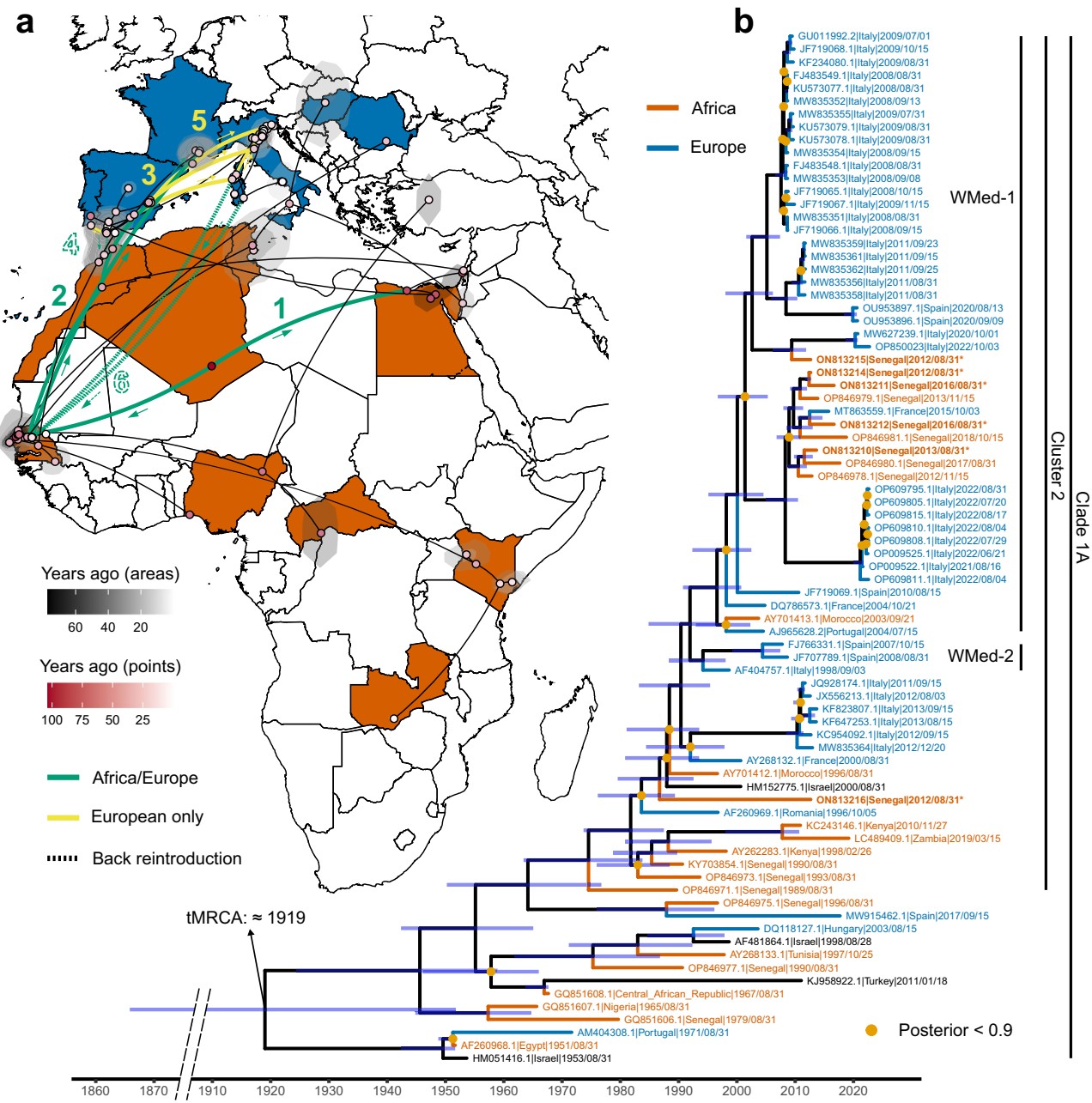

**Fig. 1 | Phylogeographic reconstruction of West Nile virus lineage 1 (WNV L1) strains. a** Geographic diffusion pattern of 80 WNV L1 genomes and their ancestors was reconstructed. African and European countries from which the samples come from are coloured in dark orange and blue, respectively. Black areas correspond to the 80% HPDs of the inferred location. The time of each area or sample is indicated by different shades (black for areas and red for dots), with the oldest samples corresponding to the most intense shade. The main events of the diffusion of the virus are indicated on the map, with numbers from 1 to 6: 1) introductions to West and East of Africa, 2) first introduction in Morocco, 2) first introduction in Morocco and then Spain, 3) movement towards France and Italy, 4) back reintroductions from Spain to Morocco and from Italy to Senegal, 5) introduction from Senegal to France in 2015, and 6) back introduction from Italy to Senegal. The direction of the spread of the virus is indicated by arrows, while colours indicate movements happened only in Africa or involving Africa and Europe (in green) and routes that took place entirely in Europe (in yellow). Dotted arcs indicate back reintroductions from Europe to Africa. **b** A molecular clock shows the phylogenetic relationships between the same 80 WNV L1 genomes. Light orange dots indicate nodes with a posterior probability <0.9, while 95% HPDs of the median ages are indicated by blue bars. African and European sequences are coloured by continent. The time of the most recent common ancestor (tMRCA) for Clade 1A was estimated to be around 1919. On the right, inside Clade 1A, three groups of interest are highlighted: cluster 2, Western Mediterranean clade 1 (WMed-1), and Western Mediterranean clade 2 (WMed-2). The map in Fig. 1 has been produced using free vector and raster map data from Natural Earth (naturalearthdata.com), downloaded from a web platform freely available at https://geojson-maps.ash.ms/.

(Fig. 1), where the oldest strain recovered appears to be the one of Egypt of 1951 (Genbank AF260968). In particular, it is likely that the African ancestor of WNV L1 Clade 1A has originated in the 1900s in a northwestern area of Africa (median=1919.02; 95% HPD: 1865.83 - 1951.67). From this area, WNV L1 of Clade 1A probably spread in two opposite directions around the mid-late 1940s, towards Western Africa (Senegal, median=1945.65; 95% HPD: 1924.41 – 1958.94) and from there to Northern Africa (Morocco, median=1987.96; 95% HPD: 1980.81 – 1993.51), and towards Eastern Africa (Egypt, median=1949.57; 95% HPD: 1942.46 – 1951.66), forming genetically related clusters and

sub-clusters that are now distributed throughout the world (Fig. 1 and Supplementary Fig. 1).

Within cluster 2, the common ancestor of the WNV L1 W-Med sub-cluster appears most likely to be located in Morocco (median=1987.96; 95% HPD: 1980.81 – 1993.51). From this area, the viral strain then spread to Spain (median=1990.39; 95% HPD: 1983.18 – 1995.4), and from there to Italy in 1998 (AF404757.1) and France in 2000 (AY268132.1), suggesting the existence of a corridor between Senegal, Morocco, and Western-Mediterranean European countries, also supported by a new introduction event from Senegal to France around 2015 (see Fig. 1). Sporadic re-introductions between these countries are also highlighted. In particular, our analysis shows i) the Moroccan strain of 2003 (AY701413.1) to have originated from an area located in South of Spain, from which the 2004 Portuguese (AJ965628.2) and French (DQ786572.1 and DQ786573.1) sequences probably also originated; and ii) two probable introductions to Senegal, around 2008 (median=2008.29; 95% HPD: 2004.97 - 2010.48) and 2012 (ON813215) from Italy, from where the virus strain seems to have arrived from the South of Spain around 2001 (median=2001.38; 95% HPD: 1996.72 – 2005.3). A sensitivity analysis performed on different subsets of our data returned similar scenarios (Supplementary Figs. 7 and 8), showing the robustness of the results.

For WNV L2, in support of what has been previously shown in the literature[21], our maximum likelihood phylogeny and molecular clock analyses of the entire dataset indicate the presence of four well-supported clades, including most African and European strains. Clade 2a includes the most genetically divergent strain of Madagascar of 1978. Clade 2b includes the oldest WNV L2 strain from South Africa, dated 1958, closely related to a Namibian strain of 2020 and to a strain from Cyprus of 1968. Clade 2c includes strains of Madagascar of 1988. Clade 2d is the largest and it is rooted in a 1958 Democratic Republic of Congo (DRC) strain (Genbank HM147824) (bootstrap support = 100) (Fig. 2 and Supplementary Fig. 2).

Clade 2d includes strains from (in chronological order) South Africa, Senegal, Uganda, CAR, Europe, and Russia (Fig. 2). Notably, we define five well-supported clusters within this clade. Cluster 1 contains the two South African strains of 1958 and 1977 and a strain of CAR of 1993. It roots Cluster 2, including strains of Russia (2018) and Iran (2017 and 2018), except for a small group of strains from Europe, such as Romania (2013 and 2014), Italy (2014), and Hungary (2017). Cluster 3 contains 13 Senegalese strains (1989 - 2006), with the exception of one strain of Ukraine (1980), and is closely related to clusters 4 and 5. Cluster 4 includes South African strains (1989, 2001, and 2008), other than one strain from Uganda (2009) and one from Namibia (2020). Finally, cluster 5 includes all Central-Southern European strains, rooted by the 2004 Hungarian sequence (Genbank MZ605382) and is rooted by cluster 4 (Fig. 2).

Our analysis evidenced the WNV L2 common ancestor to be most likely located in South Africa (Fig. 2) and to have originated between the 18th and the 19th centuries (median=1733.71; 95% HPD: 1581.10 – 1855.25). The existence of corridors between African countries is also evident between South Africa, DRC, Senegal, CAR, and Uganda (Fig. 2). Two major European introductions from Southern Africa to Hungary are also shown in 2001 (median=2001.58; 95%HPD: 1998.03 – 2003.97) and 2005 (median=2005.13; 95%HPD: 1997.65 – 2010.2): one belonging to C5, from where, following a few years of adaptation, the strain then spread to i) Austria around 2006 (median=2005.63; 95%HPD: 2003.33 – 2007.67), Greece around 2007 (median=2006.7; 95%HPD: 2004.4 – 2008.72), and Italy around 2009 (median=2009.49; 95%HPD: 2007.09 – 2010.87), and thereafter to large parts of Europe; and one belonging to C2, from where the 2014 strains of Italy and Romania originated. Also in this case, a sensitivity analysis performed on different subsets of the dataset returned a similar scenario (Supplementary Figs. 7 and 9), showing the robustness of the results.

## Discussion

Numerous studies have drawn attention to the circulation of WNV in Africa and Europe, but little is known about the genetic relationships and the introductory event dynamics of WNV L1 and L2 European and African strains[10,11,18,19]. In this study, we used phylogenetic and phylogeographic inference to uncover the origins, genetic relationships, dispersal history, and geographic patterns of most of the WNV L1 and L2 strains that have circulated and are circulating in both continents.

Of the two, WNV L1 is by far the lineage that appears to have the more complex history, with multiple clades, clusters and sub-clusters, and genetic flows among countries and continents around the world. Previous studies have described three major WNV L1 clades: 1A, which included strains from Africa, Europe, the Middle East, Asia, and America; 1B, which contained Kunjin virus, a virus circulating in Australia; and 1C, which comprised strains from India[13,14,22]. Within Clade 1A, further subclassifications of WNV L1 have been made in the past[13,23–25], with seven clusters (clusters 1 to 7) recognised, providing evidence of the very complex structure of this lineage. Our phylogenetic analysis is consistent with the previous classification and helps to provide new insights into the hierarchical structure of WNV L1 Clade 1A (Fig. 1 and Supplementary Fig. 1).

According to our analyses, the WNV strains included in most of the Clade 1A clusters (2, 3, 4, 6, and 7) are likely to originate from Africa, more specifically from West Africa. Clade 1A clusters 2, 3, 4, 6, and 7 are in fact rooted by the two ancient sequences from Nigeria and Senegal included in cluster 5 (Fig. 1 and Supplementary Fig. 1). More interestingly, cluster 2, which includes most of the European strains, appears to be rooted by the Senegal strain of 1989 (Genbank OP846971) (Fig.1). This means that the first European introduction probably originated in Senegal around the 1990s, as previously reported by Ndione et al.[15]. This hypothesis is further supported by our phylogeographic inference which not only confirms the middle of North and West Africa as the location of the common ancestor of all WNV L1 Clade 1A strains but also traces its origin back to the 19th/20th century. From this ancestral site, the virus probably spread towards Egypt and Senegal, the latter being at the origin of all WNV strains within the WMed sub-cluster (20th - 21st centuries) (Fig. 1).

The 1996 Morocco strain (Genbank AY701412) at the root of the WMed 1 and 2 subtypes implies a Moroccan origin of most of the Western-Mediterranean European WNV L1 strains. This appears to have occurred around the 1990s and reinforces the previously hypothesised assumption of the existence of a genetic flow between North of Africa (Morocco) and Italy and France [Origin and evolution of West Nile virus lineage 1 in Italy, unpublished manuscript][26].

These findings confirm the presence of a corridor between Senegal, Morocco, and Western-Mediterranean European countries, such as Portugal, Spain, France, and Italy (Fig. 1). According to our phylogenetic inference, this is not a one-way corridor as incursions of WNV strains from Europe into Africa have also been shown to occur. The WMed 1 and WMed 2 groups within the WMed sub-cluster which includes European, Moroccan (2003) and Senegalese (2012 – 2018) strains, are sister of the 2000 French strain (Fig. 1 and Supplementary Fig. 1). Our analyses indicate that the 2003 Morocco strain originated from an area located in the South of Spain, from which sequences from Portugal (2004) and France (2004) also derived. This is the first time that such an evolutionary relationship has been disclosed. The genetic similarities observed between i) the 2012-2018 Senegalese and 2015 French strains[15], all of which are closely related to the group of 2021-22 Italian sequences, and ii) the 2020-2022 Italian and 2012 Senegalese strains and those from Italy (2008 and 2011) and Spain (2020) (Fig. 1), further support this new reconstructive theory.

The evolutionary history of WNV L2 strains is much simpler to describe, with few introductory events from South Africa to Europe and no genetic flow in the opposite direction. Our maximum-likelihood phylogeny and molecular clock analysis stress the

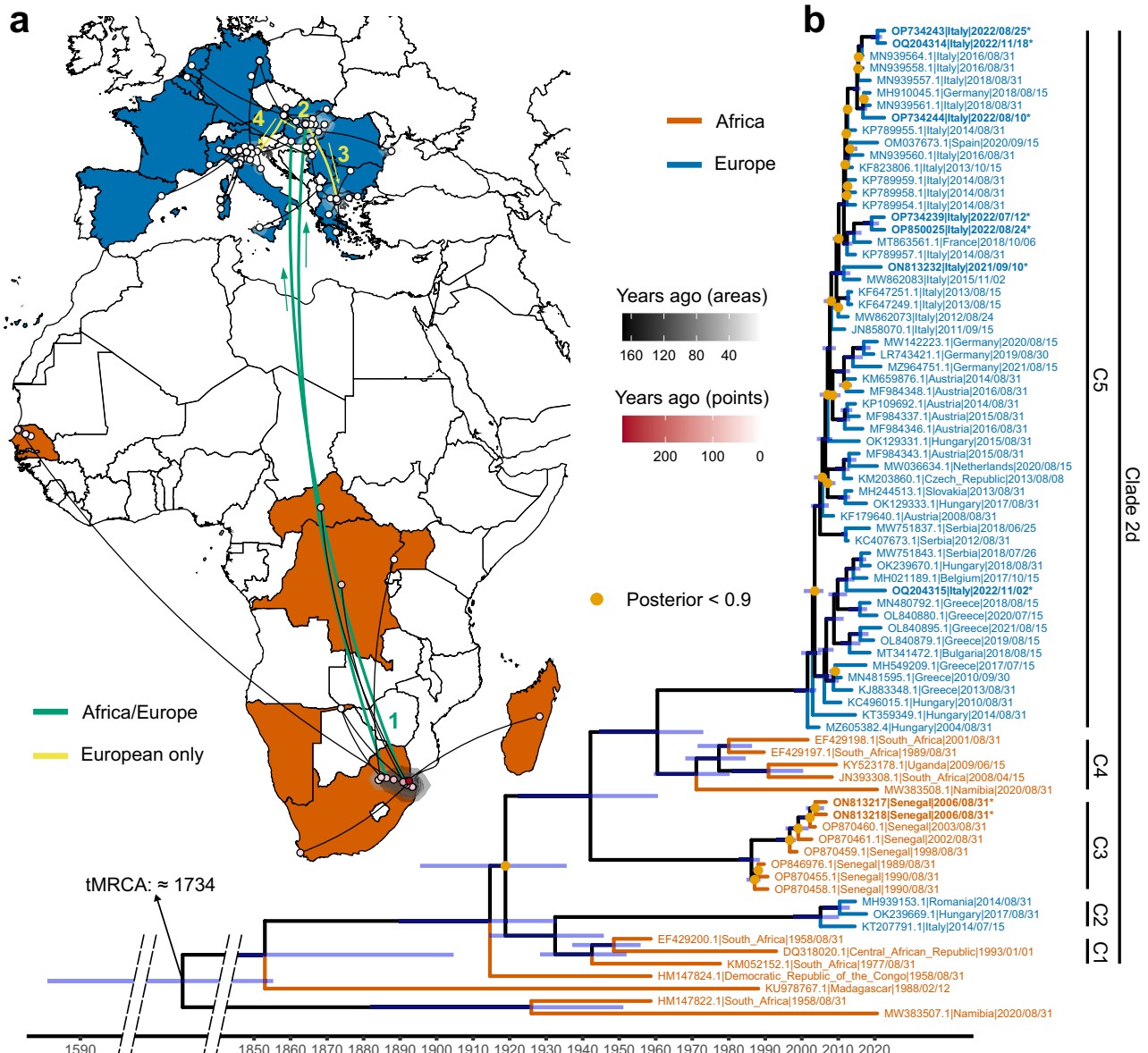

**Fig. 2 | Phylogenetic inference of West Nile virus lineage 2 (WNV L2) strains.**
**a** Geographic diffusion pattern of 80 WNV L2 genomes and their ancestors was reconstructed. African and European countries from which the samples come from are coloured in dark orange and blue, respectively. Black areas correspond to the 80%HPDs of the inferred location. The time of each area or sample is indicated by different shades (black for areas and red for dots), with the oldest samples corresponding to the most intense shade. The main events of the diffusion of the virus are indicated on the map, with numbers from 1 to 4: 1) two introductions from South Africa to Hungary in 2001 and 2005, 2) introductions from Hungary to Austria around 2006, 3) introduction from Hungary to Greece around 2007, and 4) introduction from Austria to Italy around 2009. The direction of the spread of the virus is indicated by arrows, while colours indicate movements happened only in presence of several African WNV L2 clades with common origins (Fig. 2 and Supplementary Fig. 2). Africa or involving Africa and Europe (in green) and routes that took place entirely in Europe (in yellow). Dotted arcs indicate back reintroductions from Europe to Africa. **b** A molecular clock shows the phylogenetic relationships between the same 80 WNV L2 genomes. Light orange dots indicate nodes with a posterior probability <0.9, while 95% HPDs of the median ages are indicated with blue bars. African and European sequences are coloured by continent. The time of the most recent common ancestor (tMRCA) for all WNV L2 sequences included in the analysis was estimated to be around 1733. On the right, 6 groups of interest, which are considered relevant to our discussion are highlighted: clusters 1-5 all part of Clade 2d, are indicated by C1-5 codes. The map in Fig. 2 has been produced using free vector and raster map data from Natural Earth (naturalearthdata.com), downloaded from a web platform freely available at https://geojson-maps.ash.ms/.

presence of several African WNV L2 clades with common origins (Fig. 2 and Supplementary Fig. 2). Historically, WNV L2 has circulated in Sub-Saharan Africa and Madagascar[27], before spreading to Europe[14,28]. Malagasy strains appear far apart from all the other strains included in this analysis, highlighting the presence of a local cycle probably sustained by resident birds and vector-competent mosquitoes on the island, and independent of the annual movements of migratory birds[29]. The evolution of permanent and stable local cycles, established after rare incursions into new areas, appears to be the main characteristic of the WNV L2 strains. Often, the genetic similarity of WNV L2 clades and clusters appears to be related to the specific geographical areas of circulation. This is supported by the fact that most of our WNV L2 strains cluster within the same geographical areas, as shown by i) Clade 2a and c, containing only strains from Madagascar; ii) Clade 2b, with most strains from South Africa and Namibia; and iii) Clade 2d, including cluster 1, composed of Southern and Central African strains; clusters 3 and 4, composed only of Senegal and Southern African (South Africa, Namibia, Uganda) strains, respectively; and cluster 5, with most Central-Southern European strains grouped together (Fig. 2 and Supplementary Fig. 2).

Interestingly, our phylogenetic inference shows that Clades 2c and 2d are a sister group of Clade 2b, which includes the first strain obtained in South Africa in 1958 (Supplementary Fig. 2). Furthermore, Clade 2d, which includes many African and European strains, appears to be rooted by the 1958 DRC strain, with high posterior support (Fig. 2). In particular, cluster 5 within Clade 2d including the Central-Southern European strains, all rooted by the Hungarian strain of 2004, are closely related to the South African cluster 4. These results stress the existence of a possible connection between South Africa and Europe. Our phylogeographic inference also supports this hypothesis, showing that the common ancestor of WNV L2 is likely to have originated in South Africa (Fig. 2), as already suggested in[30]. From there, WNV L2 probably spread among African countries, reaching Madagascar, DRC, CAR, Uganda, and Senegal in the 20th century, and to Europe in the 21st century. The virus, probably carried by long-distance migratory birds, was introduced into Hungary in 2004, where it was first detected, and then spread to many European countries (see Fig. 2), as previous studies have shown[14,31].

Overall, while the evolutionary history of WNV L1 is characterised by a complex behaviour possibly due to the constant connections between different continents, WNV L2 shows only one main independent introduction from Southern African countries (cluster 4) to Europe (cluster 5). Once in Europe, the lineage began to spread all over the continent, founding favourable eco-climatic conditions, which allowed it to become endemic and a serious public health concern in numerous regions and countries[28,32]. Our phylogenetic and phylogeographic reconstructions also suggest a second introduction of minor importance from Southern African countries to Europe, supported by the existence of a separate cluster 2 within Clade 2d, which includes three European strains (Italy/2014, Romania/2014, Hungary/2017) closely related to the strains of CAR (1993) and South Africa (1958 and 1977) of cluster 1.

According to these findings, WNV L1 and L2 strains have had very different eco-epidemiological and genetic evolutionary features. The more complex genomic heterogeneity of WNV L1 strains may be a consequence of the different timing of the emergence of WNV L1 and L2 strains from Africa[33], which in turn may be due to an apparently lower tendency of WNV L2 strains to spread. For this reason, WNV L2 may appear to be more conserved than WNV L1 Clade 1A strains.

Looking at the two major Eastern and Western phylogeography and genomic epidemiology of WNV L1 and L2 strains between Africa and Europe, they interestingly overlap with two most important Afro-Palaearctic bird migration flyways[34], suggesting a possible role of migratory bird species as carriers of WNV across different geographical areas. This scenario is supported by several studies showing WNV to be carried around by migratory birds[2,6,10,35], which move across the globe (https://migrationatlas.org/) spreading the virus from their original niches to new areas during this natural process[8]. In fact, during their annual cycle, birds cross the borders of numerous countries with multiple stopovers while heading to their breeding and non-breeding grounds[36]. At stopover sites, the virus can spread and become established depending on the composition of the mosquito and resident bird communities[6,37]. Each bird species appears to be characterised by a variability in migration patterns, creating a complex network of connectivity between Africa and Europe (https://migrationatlas.org/), which may underlie the enormous and variable spread of WNV L1 and L2 strains worldwide. However, it is still difficult to understand the reason why WNV L1 spreads more efficiently than L2, given that both lineages have the same chance of infecting bird species. One possible explanation could be that WNV L1 and L2 infection may have different outcomes in these species[5,38]. For instance, WNV L2 infection could result in a severe and fatal disease or in a very mild infection with brief and low levels of viraemia[39–42]. In either case, the host would decrease its capability to transmit and spread the infection.

When investigating the diverse pattern of circulation of the two viral lineages, it is important to consider the involvement of vectors in viral dispersal. In fact, vectors play a crucial role in transmitting viruses during their blood meal[43]. Most human outbreaks worldwide have been caused by L1 strains despite the strong presence of L2 in Europe and some African countries[10,15,31,32,43], and the existence of other local strains, such as L8 and KOUTV in Africa[44,45], or the Rabensburg virus and L4 in Europe[46,47]. While WNV L1 and L2 seem to have a minor impact in Africa[10], the introduction of these African strains in Europe have caused a significant number of severe WNV cases in humans and other animals[31,32,43], showing that local strains could have the ability to adapt to new environments and ecological niches and therefore be transmitted and become established. Based on the geographic area, vector species, such as mosquitoes, vary. Depending on factors such as genetic variation, environmental conditions, and evolutionary pressure, different geographic populations are characterised by a diverse behaviour and susceptibility to arbovirus infection[48]. In Africa, Gamou Fall et al. compared the L1 and L2 vector competence of African Cx. neavei and Cx. quinquefasciatus mosquitoes, often found naturally infected by WNV in Senegal, showing better transmission of L1 compared to L2[45]. In Europe, competent studies performed on Cx. pipiens mosquitoes, the major WNV competent vectors in the continent, showed these mosquitoes to be able transmitters of both L1 and L2 strains[49]. It clearly appears that the differential circulation of these two strains in Europe and Africa is correlated to multiple factors, as the ability of a mosquito to transmit the virus, the mosquito population density, the host feeding preferences, the availability of susceptible amplifying hosts, the virus divergence coming from the interaction between the virus and the avian host's immune responses; the diverse pathogenicity of circulating strains; and the association with human and animal populations[48].

Mapping lineage dispersal can be an important tool to help identify potential sources of infections, track the direction and speed of spread, and reveal areas of high transmission or disease hotspots, offering valuable information for epidemiological investigations and shedding light on the ecological and environmental processes. Our work constitutes a step in this direction. It gives an integration of extensive and diverse data obtained both from Italy and West Africa (Senegal), and reinforces prior studies and findings highlighting the genetic relationships and circulation dynamics of the two strains between European and African countries. In particular, we collected strains that are genetically distinct from the ones publicly available (e.g. ON813216|Senegal|2012/08/31* and ON813215|Senegal|2012/08/31* for WNV L1 and OQ204315|Italy|2022/11/02* for WNV L2), which allow us to improve the final phylogenetic and phylogeographic inference. By analysing the evolutionary relationships of the virus through phylogenetics and incorporating spatial information through phylogeography, we gain valuable insights into how the two viral lineages have independently evolved and dispersed across space and time. This knowledge has practical implications for surveillance and preparedness efforts concerning future outbreaks of WNV and other viruses, enhancing our ability to monitor and respond to WNV outbreaks effectively, and ultimately contributing to public health and disease control strategies.

While in this study we used a comprehensive collection of WNV L1 and L2 genomes, the limited surveillance systems of some European and African countries lead to gaps in WNV genomic surveillance data, with a limited number of newly generated genomes and sequences available mostly for certain areas and totally absent for other areas. There is still value in conducting analyses based on even larger datasets that cover regions that have been underrepresented in our current study. Expanding the dataset to include samples from these regions will enhance the robustness and reliability of future analyses, providing a more comprehensive understanding of WNV L1 and L2 dispersal dynamics and helping identify potential routes of transmission across

diverse geographic regions. Moreover, the absence of precise metadata associated with viral genomes, specifically the collection date and sample location, often unavailable or unreported, constitute a further limit. Having accurate geographic coordinates and collection dates for isolates collected over a significant period of time is essential for precisely calibrating molecular clock and phylogeographic models. This accuracy allows for dependable estimation of when and where epidemic events occurred. To address this challenge, we transformed descriptive sampling locations (like districts, provinces, or regions) into their corresponding geographic coordinates. Additionally, we addressed cases where collection dates were missing by attempting to retrieve them from the relevant research papers, if available. In situations where the collection date couldn't be identified, we assigned an estimated date. If at least the year was known, we selected the average date calculated from samples with complete information. However, it is vital to systematically ensure the availability of precise location and time metadata to allow a comprehensive phylogeographic and molecular clock analyses of viral epidemics.

Overall, the enhancement of an integrated and homogeneous surveillance plan in Europe and Africa would yield valuable benefits by allowing the generation of new consistent datasets including newly generated genomes and sequence metadata from previously unreported countries and regions. To unravel the L1 and L2 strain dynamics, it would be essential to combine these new datasets with studies on vector competence of diverse WNV strains, alongside with investigations into the abundance and host feeding preferences of various mosquito species. Finally, incorporating GPS data from migratory birds and studies on different bird species susceptibility would further contribute to the understanding on how the virus spread and which reservoirs play a significant role in its transmission across different regions, helping to mitigate the impact and predict future outbreaks, not only of this pathogen, but also of new emerging viruses.

## Methods

### Sample collection in Senegal and Italy

In Italy, sampling activities were carried out between 2001 and 2022 as part of the national surveillance plan coordinated by the Ministry of Health, the Istituto Superiore di Sanità (epidemiology and national reference laboratory, human), and the Istituto Zooprofilattico of Abruzzo and Molise (IZS-Teramo) (epidemiology and national reference laboratory, animal/entomology) (https://westnile.izs.it/j6_wnd/home, https://www.epicentro.iss.it/westnile/)[32]. In Senegal, sampling activities were carried out by the Institut Pasteur de Dakar (IPD-Dakar) in collaboration with the Ministry of Health, within the framework of the mosquito-based arbovirus surveillance system and a sentinel syndromic surveillance network (4 S). Both surveillance plans, in place since 1988[50] and 2015[51] respectively, aim to better understand the transmission dynamics of arboviruses including WNV in the country[15].

### Sample analysis

**Tissue homogenisation, viral stock preparation, RNA extraction, real-time RT-PCR.** At the IZS-Teramo, pools of avian organs and mosquito pools were homogenised in phosphate-buffered saline (PBS) with antibiotics. WNV strains were obtained from birds' internal organs or mosquito pool homogenates after one passage on *Aedes albopictus* C6/36 cell lines (ATCC Number CRL-1660™, lot number 63675525) and one to two passages on Vero monolayer cell lines (Institut Pasteur de Paris, lot number 1/99) followed by an infection[52]. Viral RNA was extracted by using the MagMAX CORE Nucleic Acid Purification KIT (Applied Biosystem, Thermo Fisher Scientific, Life Technologies Corporation, TX, USA) and amplified by multiplex real-time reverse transcription polymerase chain reactions (qRT-PCR) to detect WNV L1, WNV L2, and Usutu virus, by using the Superscript III Platinum OneStep qRT-PCR System (Invitrogen)[53].

At IPD-Dakar, arthropod and human samples derived from the WHO Collaborating Center for arboviruses and hemorrhagic fever viruses (CRORA) have been collected from the field. WNV strains were obtained after infection of C6/36 monolayer cells with homogenised mosquito pools, followed by indirect immunofluorescence assay (using in-house hyperimmune mouse ascitic fluids)[15]. To confirm WNV detection, viral RNA extraction was performed with the QIAamp viral RNA mini kit (Qiagen, Heiden, Germany) according to the manufacturer's instructions. Viral RNAs were amplified by qRT-PCR using the Quantitect Reverse Transcription Kit (Qiagen, Heiden, Germany) according to the manufacturer's instructions and a consensus WNV RT-PCR assay[54].

### Sequence retrieval and dataset preparation

In Italy and Senegal, 19 Italian WNV L2 samples, collected between 2021 and 2022, and 7 WNV L1 and 3 WNV L2 Senegalese samples, obtained between 2006 and 2016, were processed at IZS-Teramo and IPD-Dakar using next-generation sequencing (NGS) technology[15,55]. Briefly, the complete WNV genomes were obtained from RT-PCR positive samples by using the high-throughput sequencing technique in the Miseq device (Illumina, San Diego, CA, USA). The total RNA was treated with TURBO DNase (Thermo Fisher Scientific, Waltham, MA, USA) at 37 °C for 20 min, and then purified by an RNA Clean & Concentrator™-5 Kit (Zymo Research, Irvine, CA, USA). The purified RNA was used to assess the sequencing-independent single primer amplification protocol (SISPA)[15,55]. The PCR product was purified using the Molecular Biology Kit BioBasic (Biobasic inc., Markham, ON, Canada), and then quantified by using the Qubit® DNA HS Assay Kit (Thermo Fisher Scientific, Waltham, MA, USA)[55]. The sample was diluted to obtain a concentration of 100–500 ng and used for library preparation by using the Illumina DNA Prep Kit (Illumina Inc., San Diego, CA, USA) according to the manufacturer's protocol[15,55]. The resulting libraries were each identified by indexes (Nextera XT index kit V2, Illumina) and then pooled at the same concentration. Whole-genome sequencing was performed with paired-end reads using the Illumina MiSeq reagent kit v2 (300 cycles) on an Illumina MiSeq instrument, as previously described[15,55]. Assembly was automatically performed at the end of the sequencing run using the 'National Reference Centre for Whole Genome Sequencing of microbial pathogens: database and bioinformatic analysis' (GENPAT) platform, formally established at the IZS-Teramo, as described in[56]. Consensus sequences were obtained using iVar v 1.3.1[57] after mapping trimmed reads to the WNV L2 MN652880 (Greece, 2018) and WNV L1 FJ483548 (Italy, 2008) reference sequences, by using Snippy (https://github.com/tseemann/snippy).

In addition, 10 WNV L1 and 10 WNV L2 whole genome sequences, obtained from samples collected in Senegal between 1985 and 2018, were shared by the IPD-Dakar of Senegal and added to the dataset. The final dataset also included 30 WNV L1 and 45 L2 whole genome sequences, obtained at IZS-Teramo between 2008 and 2022, which were downloaded from the Supplementary Materials of[32] and [Origin and evolution of West Nile virus lineage 1 in Italy].

Using a custom R script for automated sequence retrieval, a new search was performed on the NCBI on 04/01/23, and 37 newly published worldwide WNV L1 and 39 WNV L2 sequences >= 200 nt were downloaded.

### Sequence quality-filtering and formatting

Only genomes longer than 10 Kb were retained and quality-filtering was performed[32], but no sequences were removed because they did not contain a percentage of ambiguous bases greater than 10%. A total of 229 WNV L1 and 298 WNV L2 genomes, 188 of which came from Italy (81 L1 and 97 L2) and 31 from Senegal (18 L1 and 13 L2), were selected for further analysis.

A table of sequence curated metadata is provided in Supplementary Data 1 and 2.

## Alignment, recombination detection and substitution model selection

Sequence alignment was conducted using MAFFTv7 (https://mafft.cbrc.jp/alignment/server/) with the "--auto" option and aligned sequences were trimmed using trimAlv1.2[58], selecting the "-automated1" option. Suspected recombinant sequences (the L1 sequence OP846974.1 and the L2 sequence OK239667.1) were identified with the RDP4 program[59], running the analysis under seven different methods (RDP[60], GENECONV[61], Bootscan[62], Maxchi[63], Chimaera[64], SiSscan[65] and 3Seq[66]). A sequence was considered as recombinant if detected by at least 5 methods out of 7, with a p-value threshold of 0.01 (or smaller) and excluded from the final data set (complete results for recombinant sequences are presented in Supplementary Data 4). Modelfinder program[67] was used to carry out a model selection analysis to select the most suitable substitution model for analysing the data, using parameters "-T AUTO -m TESTONLY". The best-fitting model for both the global and the down-sampled dataset used for the molecular clock (see the molecular clock and phylogeographic analysis selection below) was GTR + F + I + G4, selected according to both the Akaike Information Criterion (AIC) and the Bayesian Information Criterion (BIC).

## Maximum-likelihood phylogenies

A maximum likelihood phylogeny of the dataset, including the 228 WNV L1 and 297 L2 sequences was reconstructed by using RAxMLv8.2.12[68], with commands "-p 1989 -m GTRGAMMAI -x 2483 -# 100 -f a -T 20". Clades were annotated using the resulting topology if they had bootstrap support ≥ 90%.

## Molecular clock and phylogeographic analysis

Molecular clock analysis was performed by subselecting the sequences used for reconstructing the maximum likelihood trees, to reduce sampling bias and redundancy in the dataset, avoiding cases in which many similar genomes sampled in the same place and at the same time were present, and setting a cutoff value of 80 sequences to ensure reliable estimates in the Bayesian analysis[69]. In particular, selection was made based on genetic divergence (inferred in the maximum likelihood analysis), firstly identifying highly supported groups (bootstrap support > 90) of sequences with very low phylogenetic distance coming from the same place (or places that were close to each other) and sampled at close timepoints (i.e., the same year) and then randomly selecting sequences for each of these groups. An overview of the downsampling step, highlighting the selected and the excluded genomes for the clock analysis, is presented in Supplementary Figs. 3 and 4. We assessed the clock likeliness of our data by using TempEst v1.5.3[70], running a root-to-tip regression analysis on both the full and the reduced datasets, employing in all cases the heuristic residual mean squared method implemented in the program to test our data. We detected a strong correlation between the genetic distance and the date, which was 0.90 for the L1 clock dataset and 0.71 for the L2 clock dataset (Supplementary Data 5 and Supplementary Figs. 5 and 6), supporting the molecular clock assumption. A model selection analysis to evaluate suitable clock models and tree priors was performed on the two datasets used for molecular clocks and phylogeographic analysis. We tested the combinations of two different clock models (strict and relaxed) and two tree priors (constant coalescent and coalescent Bayesian Skyline), using two different methods for marginal likelihood estimation: path sampling and stepping-stones sampling. We tested multiple number of steps (10, 30, 50 and 100) after running the MCMC for 200*10^6 generations, selecting in the end 100 steps as the value for estimating the -log values of the marginal likelihoods (both for the L1 and the L2 datasets). We used the obtained values to calculate Bayes factors to identify the most suitable model for analysing our data. For both datasets, a combination of relaxed clock and

coalescent Bayesian Skyline resulted as the best one among all the ones tested, with a log BF > 5 (Supplementary Data 6), which strongly supports the selected model over all the others tested[71]. Phylogeography was reconstructed by using continuous traits (latitudinal and longitudinal coordinates for each sequence; full list in Supplementary Data 3) in BEASTv1.10.4[72]. The analysis was divided in two different partitions, one for sequence data and the other for the continuous coordinates. For the first partition, we employed a GTR substitution model (selected in the previous steps) with a gamma distribution and 4 categories. For the second partition, a Cauchy RRW substitution model was selected, with bivariate traits representing latitude and longitude and adding random jitter to the tips (jitter window size: 0.01), reconstructing states for all ancestors. We ran the analysis with the clock and tree priors that resulted as most suitable to our data after a model selection (relaxed clock with a log-normal distribution and coalescent Bayesian Skyline tree prior), setting an MCMC length of 1 billion generations and sampling every 1*10^5 steps. Convergence was assessed using Tracer V1.7.1[73], ensuring that all parameters were above a significance threshold of ESS (> 200). Maximum-clade credibility trees were obtained using TreeAnnotator[72], with 15% burnin, and median heights. Spatio-temporal patterns of WNV evolution were visualised using SpreaD3[74]. GeoJSON shapefiles of the maps in the plots were downloaded at https://geojson-maps.ash.ms/, a web interface containing maps sourced from Natural Earth and freely available (https://www.naturalearthdata.com/about/terms-of-use/). To test how choosing a different set of genomes may have influenced our results, we set up a sensitivity analysis by randomly sampling sequences from our molecular clock datasets. We built 6 different datasets by randomly sampling with a custom script 75%, 50% and 30% of our two datasets used for molecular clock analysis and ran the analyses for each dataset using the same calibrations but setting a different MCMC length (250*10^6 generations sampling every 2.5*10^4 steps and 200*10^6 generations sampling every 2*10^4 steps for the 30% subsample).

## Ethical statement

The cases reported in this study were investigated with routine procedures according to the national surveillance plan for arbovirus infections. Therefore, no approval was required from the ethics committee.

## Reporting summary

Further information on research design is available in the Nature Portfolio Reporting Summary linked to this article.

# Data availability

All data that support the findings of this study have been deposited in Figshare database under the project no 160822. Particularly, alignments used for the phylogenetic analysis, model selection analysis, and tree files can be found at https://doi.org/10.6084/m9.figshare.22182418, while phylogeographic inference tree files, geographic coordinates, and videos can be found at https://doi.org/10.6084/m9.figshare.23660571. Accession codes of all used sequences are provided in Supplementary Data 1 and 2. Raw data for Supplementary Figures 5–7 are provided in Source Data. Source data are provided with this paper.

# Code availability

Using a custom R script for automated sequence retrieval, a new search was performed on the NCBI on 04/01/23, and 37 newly published worldwide WNV L1 and 39 WNV L2 sequences >= 200 nt were downloaded. The code to retrieve and filter sequences is available at https://github.com/andrea-silverj/WNV-Afr_Eur/tree/main/scripts (https://zenodo.org/badge/latestdoi/611814462).

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

## Acknowledgements

The authors acknowledge all personnel in the lab and the field for their skill and technical support. Special thanks to the WHO Collaborating Center for arboviruses and Hemorrhagic Fever Viruses (CRORA) in the Institut Pasteur de Dakar, and in particular to Arame Ba, Moussa Dia, Oumar Ndiaye, Diogop Camara and Chérif Sylla for their constant work and support during the diagnostic activities. This research has been funded by the Italian Ministry of Health, the law on 19 January 2001. This research has been partially supported by the Italian Ministry of Health research grant no. MSRCTE 04/19, and by the EU grant 874850 MOOD and is catalogued as MOOD 069. This research has been sup-ported by an international Ph.D. initiative including Fondazione Edmund Mach, University of Trento, and Istituto Zooprofilattico of Teramo. The contents of this publication are the sole responsibility of the authors and don't necessarily reflect the views of the European Commission.

## Author contributions

Conceptualization, G.Men., M.H.D.N., M.M.D., A.R., R.R., O.F., F.M., and G.S.; methodology, G.Men., A.S., M.H.D.N., and G.S.; validation, F.M., A.S., G.Men., M.H.D.N., M.M.D., O.F., A.R., and G.S.; formal analysis, A.S., and G.Men.; investigation, L.T., A.L., V.C., M.D.D., A.P., M.H.D.N., R.M., D.D., M.A., B.S., V.D.L., M.M.D., M.D., A.G., E.H.N., and G.Men.; resources, G.S., A.R., and O.F.; data curation, A.S., V.C., I.M., A.B., M.M.D., M.H.D.N., and G.Men.; writing G.Men.; original draft preparation, G.Men., M.H.D.N., A.S., and G.S.; writing—review and editing, F.M., M.H.D.N., A.R., O.F., M.M.D., O.R.S., and G.S.; visualization, A.S., G.Men., L.T., A.L., I.M., A.P., A.B., D.D., V.C., M.H.D.N., M.D.D., M.A., R.M., B.S., V.D.L., M.M., C.C., G.Mar., R.R., F.M., N.S., M.D., A.G., E.H.N, G.F., A.R., O.R.S., O.F., and G.S.; supervision, O.R.S., F.M., A.R., O.F., M.H.D.N., M.M.D., and G.S.; project administration, G.S.; funding acquisition, G.S. All authors have read and agreed to the published version of the manuscript.

## Competing interests

The authors declare no competing interests.
