## [Peer Review File · Nature Communications]

Spatial and temporal dynamics of West Nile virus between Africa and EuropeREVIEWER COMMENTS

Reviewer #1 (Remarks to the Author):

Review of Mencattelli et al:

The authors provide further evidence for the circulation of WNV between Africa and Europe based on extensive integration of geographic, genomic, phylogenetic and phylo-geographic data, highlighting WBV Lineages 1 and 2. They found cross-circulation for L1 (a) and a one-way introduction for L2. They also associated the overlap of these circulations with major bird flyways.

The data analysis, interpretation and conclusion appear solid (though this reviewer is not an expert on genetic and molecular analyses), and the work does support the claim. There are no major flaws in the data analysis, interpretation or conclusions. The methodologies, which I am qualified to evaluate appear sound, and there is enough detail provided in the methods for the work to be reproduced.

The main value of the work is in its comprehensiveness and integration of extensive and diverse data obtained both from Italy and from West Africa (Senegal). It builds on several earlier papers led by the first author on the introduction and circulation of lineages 1 and 2 in Italy (e.g., Mencatelli et al. *Viruses*. 2023, Mencateli et al. *TMID* 2022; both of them are cited in this paper).

The results reinforce prior studies and findings, and highlight the genetic relationships and circulation dynamics of WNV lineages 1 and 2 between Italy and sub-Saharan and northern Africa. The combined use of phylogenetics and phylogeography is a strength of the study.

The complex history of L1, and in particular, Clade 1A is a key finding of the study, in contrast to the simpler history of L2. The overlap of WNV phylogeography and genomic epidemiology is well documented from various other studies worldwide, in particular some from Israel (e.g., Schwartz et al 2022). In general more references to some of these other studies, including also, for example Lustig et al 2016 would be worthwhile, in order to put

the study in better perspective with regard to the global phylogenetic/phylogeography of WNV

The work is of significance to the field and related fields, and adds depth to existing literature. The work does support the conclusions and claims. While the results are not surprising, they are of value both for our understanding of WNV eco-epidemiology and for surveillance and preparation for future outbreaks of WNV and other viruses.

References mentioned:

Schwartz G, Tirosh-Levy S, Bider S, Lublin A, Farnoushi Y, Erster O, Steinman A. West Nile Virus in Common Wild Avian Species in Israel. *Pathogens*. 2022 Jan 17;11(1):107. doi: 10.3390/pathogens11010107. PMID: 35056055; PMCID: PMC8780237.

Yaniv Lustig, Musa Hindiyeh, Laor Orshan, Leah Weiss, Ravit Koren, Shiri Katz-Likvornik, Hila Zadka, Aharona Glatman-freedman, Ella Mendelson, Lester M. Shulman, Mosquito Surveillance for 15 Years Reveals High Genetic Diversity Among West Nile Viruses in Israel, *The Journal of Infectious Diseases*, Volume 213, Issue 7, 1 April 2016, Pages 1107–1114, <https://doi.org/10.1093/infdis/jiv556>

Reviewer #2 (Remarks to the Author):

Mencattelli et al. used whole genome sequencing and phylogeographic analysis to investigate the transmission dynamics of West Nile virus (WNV) across Africa and Europe. Their research indicates that WNV L1 lineage originated from Northern-Western Africa, with back-and-forth diffusion between West Africa and Southern-Western Europe whereas L2 originated from Southern Africa, with one main introduction from South Africa to Europe and no back introductions detected. The study has important implications for our understanding of origins and circulating patterns of WNV lineages across two continents.

Below are my specific comments, hoping to improve this work/manuscript:

1. Line 170-172 Recombination analysis: it was mentioned that potential recombination was

detected using RDP4 program, readers would be benefited from some more details about which methods, criteria and p-value threshold etc were used.

2. Line 183-186 Molecular clock analysis: it seems the authors have not check how well the temporal signal (i.e. "clockness") of WNV evolution is. It would be useful to perform a root-to-tip regression to do some preliminary assessment on the clockness. Formal statistical testing (Bayes factor etc) and identification of appropriate clock model is highly recommended.

3. Line 183-186: It would be clearer to readers by providing more details about subselection criteria used in generating the reduced data set for clock and phylogeographic analysis. The details would help to justify the representativeness of the data subset, and any potential bias to the analysis.

4. Line 192: Could the authors do some analysis to show sampling bias are not influencing the conclusion? Over-or undersampling of sequences from a specific site can have a significant influence on the estimated ancestral locations since the ancestral reconstruction of locations depends on the availability of data. In this case, a sensitivity analysis can be done by repeating the analysis with different subsets of sequences and comparing the results.

5. Figure 1a,2a: Could authors label/highlight important locations and migration events on the map to make it easier to pick up major results?

6. Figure 1b,2b: it will be easier to understand if sequences are colored by different places and are described in the legend; Will be more informative by highlighting which sequences are generated by this work; The yellow dot label is confusing – should it be >0.6 ? By the way, clade probability of over 0.9 or over 0.95 is more commonly used as the cut-off of well-supported topology.

7. Line 316: Discussion and conclusions: apart from summarizing findings and the consistency with previous studies, authors are suggested to include the acknowledgement

of limitations in this study, such as methodological issues, data quality or factors that may influence the interpretation of results.

8. Line 30: Better more consistently use L1 instead of L1a.

9. It would be helpful to highlight the uniqueness of this work by explaining the limit of existing WNV data available, and how the new WNV genomes sequenced by this study could uniquely provide the insights/opportunity to assess such hypothesis.

Tommy Lam

Reviewer #3 (Remarks to the Author):

This is a very well and clearly written manuscript. I am not a specialist in Genomic analysis, so cannot really comment effectively on the methodologies or any specific aspects of it, but as this technology is fairly well established by now, I doubt there are any major methodological comments needed.

My one substantive comment concerns the vectors - in the discussion you speculate that the difference in spread pattern/rate between L1 and L2 may be to do with the pathogens impact on bird hosts. I wonder whether the vector competence for the two strains varies significantly, and whether this might also have a differential impact on each lineage

My recommendation is to publish with minor revisions

Reviewer's major & minor comments:

Reviewer #1:

“The authors provide further evidence for the circulation of WNV between Africa and Europe based on extensive integration of geographic, genomic, phylogenetic and phylogeographic data, highlighting WNV Lineages 1 and 2. They found cross-circulation for L1 (a) and a one-way introduction for L2. They also associated the overlap of these circulations with major bird flyways. The data analysis, interpretation and conclusion appear solid (though this reviewer is not an expert on genetic and molecular analyses), and the work does support the claim. There are no major flaws in the data analysis, interpretation, or conclusions. The methodologies, which I am qualified to evaluate appear sound, and there is enough detail provided in the methods for the work to be reproduced. The main value of the work is in its comprehensiveness and integration of extensive and diverse data obtained both from Italy and from West Africa (Senegal). It builds on several earlier papers led by the first author on the introduction and circulation of lineages 1 and 2 in Italy (e.g., Mencattelli et al. *Viruses*. 2023, Mencattelli et al. *TMID* 2022; both of them are cited in this paper). The results reinforce prior studies and findings and highlight the genetic relationships and circulation dynamics of WNV lineages 1 and 2 between Italy and sub-Saharan and northern Africa. The combined use of phylogenetics and phylogeography is a strength of the study. The complex history of L1, and in particular, Clade 1A is a key finding of the study, in contrast to the simpler history of L2. The overlap of WNV phylogeography and genomic epidemiology is well documented from various other studies worldwide, in particular some from Israel (e.g., Schwartz et al 2022). In general, more references to some of these other studies, including also, for example Lustig et al 2016 would be worthwhile, in order to put the study in better perspective with regard to the global phylogenetic/phylogeography of WNV. The work is of significance to the field and related fields and adds depth to existing literature. The work does support the conclusions and claims. While the results are not surprising, they are of value both for our understanding of WNV eco-epidemiology and for surveillance and preparation for future outbreaks of WNV and other viruses.

References mentioned:

1. Schwartz G, Tirosh-Levy S, Bider S, Lublin A, Farnoushi Y, Erster O, Steinman A. West Nile Virus in Common Wild Avian Species in Israel. Pathogens. 2022 Jan 17;11(1):107. doi: 10.3390/pathogens11010107. PMID: 35056055; PMCID: PMC8780237.

2. Yaniv Lustig, Musa Hindiyeh, Laor Orshan, Leah Weiss, Ravit Koren, Shiri Katz-Likvornik, Hila Zadka, Aharon Glatman-freedman, Ella Mendelson, Lester M. Shulman, Mosquito Surveillance for 15 Years Reveals High Genetic Diversity Among West Nile Viruses in Israel, The Journal of Infectious Diseases, Volume 213, Issue 7, 1 April 2016, Pages 1107–1114, <https://doi.org/10.1093/infdis/jiv556>”

We thank the reviewer for the appreciation of our work.

We believe the two references:

1. “Schvartz G, Tirosh-Levy S, Bider S, Lublin A, Farnoushi Y, Erster O, Steinman A. West Nile Virus in Common Wild Avian Species in Israel. Pathogens. 2022 Jan 17;11(1):107. doi: 10.3390/pathogens11010107. PMID: 35056055; PMCID: PMC8780237.”
2. “Yaniv Lustig, Musa Hindiyeh, Laor Orshan, Leah Weiss, Ravit Koren, Shiri Katz-Likvornik, Hila Zadka, Aharon Glatman-freedman, Ella Mendelson, Lester M. Shulman, Mosquito Surveillance for 15 Years Reveals High Genetic Diversity Among West Nile Viruses in Israel, The Journal of Infectious Diseases, Volume 213, Issue 7, 1 April 2016, Pages 1107–1114, <https://doi.org/10.1093/infdis/jiv556>”

are significant, putting the study in a better perspective regarding the global WNV phylogenetics and phylogeography. For this reason, we revised our manuscript adding the two references to our work.

Particularly, the reference “Schvartz, G. et al. West Nile Virus in Common Wild Avian Species in Israel. Pathogens 11, 107 (2022)” is now classified in the text as reference n.6. It has been added to:

- New Line 50: “Although in the majority of the susceptible bird species infection generally does not lead to severe clinical signs, in some species it might cause severe neurological signs, including death⁴⁻⁶.”
- New line 357: “This assumption is confirmed by several studies showing WNV to be carried around by migratory birds^{2,6,10,35}, which move across the globe

(<https://migrationatlas.org/>) spreading the virus from their original niches to new areas during this natural process ⁸.”

- New Line 361: “At stopover sites, the virus can spread and become established depending on the composition of the mosquito and resident bird communities ^{6,37}”.

Also, the reference “Lustig, Y. et al. Mosquito Surveillance for 15 Years Reveals High Genetic Diversity Among West Nile Viruses in Israel. *J Infect Dis* 213, 1107–1114 (2016)” is now classified in the text as ref. n. 13 and it has been added to the text to:

- New Line 74: “At least nine lineages have been characterised worldwide, four of which circulate in Africa [lineage 1 (L1 or L1a), lineage 2 (L2), lineage 7 (L7 - now classified as the Koutango virus), and putative lineage 8 (L8)] ^{10,13}.”
- New line 86: “Despite the existence of several studies focusing on WNV circulation ^{10,11,13,18,19}, little is known about the genetic relationships between European and African WNV L1 and L2 strains.”
- New line 265: “Previous studies have described three major WNV L1 clades: 1A, which included strains from Africa, Europe, the Middle East, Asia, and America; 1B, which contained Kunjin virus, a virus circulating in Australia; and 1C, which comprised strains from India ^{13,14,22}”. Within clade 1A, further subclassifications of WNV L1 have been made in the past ^{13,23–25}”.

Reviewer #2:

“Mencattelli et al. used whole genome sequencing and phylogeographic analysis to investigate the transmission dynamics of West Nile virus (WNV) across Africa and Europe. Their research indicates that WNV L1 lineage originated from Northern-Western Africa, with back-and-forth diffusion between West Africa and Southern-Western Europe whereas L2 originated from Southern Africa, with one main introduction from South Africa to Europe and no back introductions detected. The study has important implications for our understanding of origins and circulating patterns of WNV lineages across two continents. Below are my specific comments, hoping to improve this work/manuscript:

1. Line 170-172 Recombination analysis: it was mentioned that potential recombination was detected using RDP4 program, readers would be benefited from some more details about which methods, criteria and p-value threshold etc were used.

To detect recombinants in our dataset, we ran the RDP4 program, employing 7 different methods: RDP, GENECONV, Bootscan, Maxchi, Chimaera, SiSscan, and 3Seq. A sequence was considered recombinant if detected by at least 5 methods out of 7, with a p-value threshold of 0.01 (or smaller). We added a small table in the supplementary materials (Supplementary Table 4), reporting results for the sequences detected as recombinants. We added these details to the methods section.

We modified the text, as follows (Now lines 521-526):

“Suspected recombinant sequences (the L1 sequence OP846974.1 and the L2 sequence OK239667.1) were identified with the RDP4 program⁵⁹, running the analysis under seven different methods (RDP⁶⁰, GENECONV⁶¹, Bootscan⁶², Maxchi⁶³, Chimaera⁶⁴, SiSscan⁶⁵ and 3Seq⁶⁶). A sequence was considered as recombinant if detected by at least 5 methods out of 7, with a p-value threshold of 0.01 (or smaller) and excluded from the final data set (complete results for recombinant sequences are presented in Supplementary Table 4).”

References:

59. Martin, D. P., Murrell, B., Golden, M., Khoosal, A. & Muhire, B. RDP4: Detection and analysis of recombination patterns in virus genomes. *Virus Evol* **1**, vev003 (2015).

60. Martin D. Rybicki E. (2000) ‘RDP: Detection of Recombination Amongst Aligned Sequences’, *Bioinformatics*, 16: 562–3.

61: Padidam M. Sawyer S. Fauquet C. M. (1999) ‘Possible Emergence of New Geminiviruses by Frequent Recombination’, *Virology*, 265: 218–25.

62: Salminen M. O. et al. . (1995) ‘Identification of Breakpoints in Intergenotypic Recombinants of HIV Type 1 by BOOTSCANning’, *AIDS Research and Human Retroviruses*, 11: 1423–5.

63. Maynard Smith J. (1992) ‘Analyzing the Mosaic Structure of Genes’, *Journal of Molecular Evolution*, 34: 126–9.

64. Posada D. Crandall K. A. (2001) 'Evaluation of Methods for Detecting Recombination from DNA Sequences: Computer Simulations', Proceedings of the National Academy of Sciences of the United States of America, 98: 13757–62.
65. Gibbs M. J. Armstrong J. S. Gibbs A. J. (2000) 'Sister-Scanning: A Monte Carlo Procedure for Assessing Signals in Recombinant Sequences', Bioinformatics, 16: 573–82.
66. Boni M. F. Posada D. Feldman M. W. (2007) 'An Exact Nonparametric Method for Inferring Mosaic Structure in Sequence Triplets', Genetics, 176: 1035–47.

2. Line 183-186 Molecular clock analysis: it seems the authors have not checked how well the temporal signal (i.e. “clockness”) of WNV evolution is. It would be useful to perform a root-to-tip regression to do some preliminary assessment on the clockness. Formal statistical testing (Bayes factor etc) and identification of appropriate clock model is highly recommended.

We thank the reviewer for this comment. We further tested the clocklikeness of our data by performing a root-to-tip regression analysis using TempEst v1.5.3. We tested the signal of 4 different datasets: all WNV L1 sequences together, all WNV L2 together and the WNV L1 and WNV L2 reduced datasets used for the clock, employing in all cases the heuristic residual mean squared method implemented in the program to test our data, best-fitting the root of the trees. We detected a strong correlation between the genetic distance and the date, which was 0.90 for the L1 clock dataset and 0.71 for the L2 clock dataset. These values change only slightly in the reduced versions of the datasets. This denotes a clear and robust clock signal in our data, which justifies the assumption of molecular clock. We added this section to our methods and provided a table showing all results (Supplementary Table 5 and Supplementary Figures 5-6).

We modified the text, as follows (Now lines 549-554):

“We assessed the clocklikeness of our data by using TempEst v1.5.3⁷⁰, running a root-to-tip regression analysis on both the full and the reduced datasets, employing in all cases the heuristic residual mean squared method implemented in the program to test our data. We detected a strong correlation between the genetic distance and the date, which was 0.90 for the L1 clock

dataset and 0.71 for the L2 clock dataset (Supplementary Table 5 and Supplementary Figures 5-6), supporting the molecular clock assumption.”

A model selection analysis was performed on the two datasets used for molecular clocks and phylogeographic analysis. We tested the combinations of two different clock models (strict and relaxed) and two tree priors (constant coalescent and Bayesian Skyline) using two different methods for marginal likelihood estimation: path sampling and stepping-stones sampling. We estimated the -log values of the marginal likelihoods to calculate Bayes factors, to identify the most suitable model for analysing our data. A combination of relaxed clock and Bayesian Skyline resulted as the best one among all the ones tested, with a log BF > 5, which strongly supports the selected model over all the others tested. We put the table with the estimated marginal likelihoods and the log Bayes factors in Supplementary Table 6. Therefore, we repeated the analysis with the new set of priors, running the MCMC chain for 1 billion generations. Results under the new priors are not much different from the previous ones, with changes that affected more the time estimates than other parameters.

We modified the text, as follows (Now lines 554-566):

“A model selection analysis to evaluate suitable clock models and tree priors was performed on the two datasets used for molecular clocks and phylogeographic analysis. We tested the combinations of two different clock models (strict and relaxed) and two tree priors (constant coalescent and coalescent Bayesian Skyline), using two different methods for marginal likelihood estimation: path sampling and stepping-stones sampling. We tested multiple number of steps (10, 30, 50 and 100) after running the MCMC for 200×10^6 generations, selecting in the end 100 steps as the value for estimating the -log values of the marginal likelihoods (both for the L1 and the L2 datasets). We used the obtained values to calculate Bayes factors to identify the most suitable model for analysing our data. For both datasets, a combination of relaxed clock and coalescent Bayesian Skyline resulted as the best one among all the ones tested, with a log BF > 5 (Supplementary Table 6), which strongly supports the selected model over all the others tested⁷¹.”

References:

70. Rambaut, Lam, de Carvalho & Pybus (2016) Exploring the temporal structure of heterochronous sequences using TempEst. *Virus Evolution* 2: vew007 DOI: <http://dx.doi.org/10.1093/ve/vew007>

71. Kass, R. E. & Raftery, A. E. Bayes Factors. *Journal of the American Statistical Association* 90, 773–795 (1995).

3. Line 183-186: It would be clearer to readers by providing more details about subselection criteria used in generating the reduced data set for clock and phylogeographic analysis. The details would help to justify the representativeness of the data subset, and any potential bias to the analysis.

The criteria used for the sub-selection of sequences, as suggested by Reviewer 2, is indeed very relevant, so we added more details to clarify this point. We downsampled our datasets considering 3 different factors: 1) the results of the maximum likelihood phylogeny, 2) the collection date and 3) the location of the samples from which the sequences were obtained. As there was a sampling bias in the dataset, with cases in which many similar sequences sampled in the same place and at the same time were present, we applied the following procedure to reduce the number of nearly-identical sequences:

- 1) Identify highly supported groups of sequences (bootstrap support > 90) with very low phylogenetic distance (inferred from the tree), coming from the same place (or places that were close to each other) and sampled at close timepoints (i.e., the same year).
- 2) Randomly select sequences in the identified groups.
- 3) Keep cutting the dataset until reaching a threshold of 80 sequences, to ensure reliable estimates for all parameters in the Bayesian analysis (Poon et al., 2012).

We now provide an overview of the downsampling step, highlighting the selected and the excluded genomes in Supplementary Figures 3-4.

This approach was a mix between a manual curation (which can be assisted by tools like TreePruner (Cit. Krishnamoorthy, M., Patel, P., Dimitrijevic, M. et al. Tree pruner: An efficient tool for selecting data from a biased genetic database. *BMC Bioinformatics* 12, 51 (2011). <https://doi.org/10.1186/1471-2105-12-51>) and an automatic approach (e.g. Treemmer, Cit. Menardo, F., Loiseau, C., Brites, D. et al. Treemmer: a tool to reduce large phylogenetic

datasets with minimal loss of diversity. *BMC Bioinformatics* 19, 164 (2018). <https://doi.org/10.1186/s12859-018-2164-8>), as we established a set of well-defined rules to prune our dataset. Automatic approaches must be preferred in the case of big datasets, but they have their own pitfalls and not many include the geographic information and the collection date in the process of selection. In general, it is always a good practice to do a final check manually, especially in the presence of datasets that are not too large.

Setting a threshold for the size of our dataset was also a way to reduce the sampling bias in our analysis, avoiding redundancy, maximising the phylogenetic signal (sequences that are very similar to each other contain very low information) and reducing the computational burden (similarly to Poon et al., 2012). Biases are still present though, as many regions of Europe and Africa have never been sampled or have been neglected across the years, but this doesn't depend on this study. We talk more in detail about this point in the final part of the discussion of the article, highlighting the importance of expanding sampling and sequencing efforts to more areas in the future, especially in Africa.

We modified the text, as follows (Now lines 539-549):

“Molecular clock analysis was performed by subselecting the sequences used for reconstructing the maximum likelihood trees, to reduce sampling bias and redundancy in the dataset, avoiding cases in which many similar genomes sampled in the same place and at the same time were present, and setting a cutoff value of 80 sequences to ensure reliable estimates in the Bayesian analysis ⁶⁹. In particular, selection was made based on genetic divergence (inferred in the maximum likelihood analysis), firstly identifying highly supported groups (bootstrap support > 90) of sequences with very low phylogenetic distance coming from the same place (or places that were close to each other) and sampled at close timepoints (i.e., the same year) and then randomly selecting sequences for each of these groups. An overview of the downsampling step, highlighting the selected and the excluded genomes for the clock analysis, is presented in Supplementary Figures 3-4.”

Reference:

69. Poon, A. F. Y. et al. Reconstructing the Dynamics of HIV Evolution within Hosts from Serial Deep Sequence Data. *PLoS Comput Biol* 8, e1002753 (2012).

4. Line 192: Could the authors do some analysis to show sampling bias are not influencing the conclusion? Over-or undersampling of sequences from a specific site can have a significant influence on the estimated ancestral locations since the ancestral reconstruction of locations depends on the availability of data. In this case, a sensitivity analysis can be done by repeating the analysis with different subsets of sequences and comparing the results.

To test how choosing a different set of genomes may have influenced our results, we set up a sensitivity analysis by randomly sampling sequences from our molecular clock datasets.

We built 6 different datasets by randomly sampling 75%, 50%, and 30% of our two datasets used for molecular clocking and ran the analyses for each dataset using the same calibrations but setting a different MCMC length (250*10⁶ generations sampling every 2.5*10⁴ steps and 200*10⁶ generations sampling every 2*10⁴ steps for the 30% subsample). Results are shown in Supplementary Figures 7-9 and are all consistent, returning similar phylogeographic scenarios. This further indicates the robustness of our datasets and the presence of a clear signal which is not heavily affected by sample size and sampling biases.

We modified the text, as follows (Now lines 582-588):

“To test how choosing a different set of genomes may have influenced our results, we set up a sensitivity analysis by randomly sampling sequences from our molecular clock datasets. We built 6 different datasets by randomly sampling 75%, 50% and 30% of our two datasets used for molecular clocking and ran the analyses for each dataset using the same calibrations but setting a different MCMC length (250*10⁶ generations sampling every 2.5*10⁴ steps and 200*10⁶ generations sampling every 2*10⁴ steps for the 30% subsample).”

5. Figure 1a,2a: Could authors label/highlight important locations and migration events on the map to make it easier to pick up major results?

We highlighted important locations and migration events by annotating the map with different colours, drawing arrows to show the direction of the movements (see the captions of Figures 1 and 2). In particular, we highlighted the following main events:

- For WNV L1:

1. Introductions to West (Senegal, median=1945.65; 95% HPD: 1924.41 – 1958.94) and East (Egypt, median=1949.57; 95% HPD: 1942.46 – 1951.66) of Africa
2. First introduction in Morocco (median=1987.96; 95%HPD: 1980.81 – 1993.51) and then in Spain (median=1990.39; 95%HPD: 1983.18 – 1995.4): Africa/Europe corridor
3. Movement towards Italy (1998) and France (2000)
4. Back reintroductions, from Spain to Morocco in 2003 and from Italy to Senegal in 2009 and 2012
5. Introduction from Senegal to France in 2015.

- For WNV L2:

1. Two introductions from South Africa to Hungary in 2001 (median=2001.58; 95%HPD: 1998.03 – 2003.97) and 2005 (median=2005.13; 95%HPD: 1997.65 – 2010.2)
2. Introduction from Hungary to Austria around 2006 (median=2005.63; 95%HPD: 2003.33 – 2007.67)
3. Introduction from Hungary to Greece around 2007 (median=2006.7; 95%HPD: 2004.4 – 2008.72)
4. Introduction from Austria to Italy in 2009 (median=2009.49; 95%HPD: 2007.09 – 2010.87).

6. Figure 1b,2b: it will be easier to understand if sequences are colored by different places and are described in the legend; Will be more informative by highlighting which sequences are generated by this work; The yellow dot label is confusing – should it be >0.6? By the way, clade probability of over 0.9 or over 0.95 is more commonly used as the cut-off of well-supported topology.

We coloured sequences from Africa in orange and European sequences in blue. We think this choice maximises the readability of the figure and puts in evidence the dynamics of viral circulation between these two continents. We avoided colouring sequences simply by country, as this would have meant to use more than 15 different colours, making the figure very difficult to read. We think using colours was a very good suggestion, as it is now more evident to appreciate how the new sampling allowed us to explore the hypothesis of WNV circulation between Africa and Europe. We marked the sequences produced in this study with the “*” symbol and bold fonts. Sequences from Italy were all collected and sequenced by the IZS, while sequences from Senegal were collected from the IPD and partly sequenced at IZS. As we had only a few nodes with very low posterior probabilities, we decided to highlight those ones by colouring them. Following reviewer’s 2 suggestions, we coloured nodes with posterior probability below 0.9 (as most of the nodes have a posterior higher than that, colouring nodes with high posteriors would have reduced the readability of the figure, with too many shapes on the tree). Similarly, we changed the threshold for highlighting nodes for trees shown in Supplementary Figures 1 and 2, colouring nodes with bootstrap support < 90.

7. Line 316: Discussion and conclusions: apart from summarizing findings and the consistency with previous studies, authors are suggested to include the acknowledgement of limitations in this study, such as methodological issues, data quality or factors that may influence the interpretation of results.

We thank the reviewer for these important suggestions that we believe greatly improved our work. We added the limitation of our study to new lines 411-432, as follows:

“While in this study we used a comprehensive collection of WNV L1 and L2 genomes, the limited surveillance systems of some European and African countries lead to gaps in WNV genomic surveillance data, with a limited number of newly generated genomes and sequences available mostly for certain areas and totally absent for other areas. There is still value in conducting analyses based on even larger datasets that cover regions that have been underrepresented in our current study. Expanding the dataset to include samples from these regions will enhance the robustness and reliability of our analyses, providing a more comprehensive understanding of WNV L1 and L2 dispersal dynamics and helping identify potential routes of transmission across diverse geographic regions. Moreover, the absence of

precise metadata associated with viral genomes, specifically the collection date and sample location, often unavailable or unreported, constitute a further limit. Having precise geographic coordinates and collection dates for isolates gathered over a substantial timeframe is crucial for the accurate calibration of molecular clock and phylogeographic models, enabling reliable estimation of the timing and location of epidemic events. We overcame this limitation by converting into corresponding geographic coordinates the descriptive sampling location, such as district, province, or region. Moreover, we checked the cases lacking in information regarding the collection date, attempting to retrieve it from the corresponding paper, if available. When unable to identify the collection date, we put a putative date, if at least the year was known, choosing the average date calculated from the samples with complete information. However, it is vital to systematically ensure the availability of precise location and time metadata to allow a comprehensive phylogeographic and molecular clock analysis of viral epidemics.”

8. Line 30: Better more consistently use L1 instead of L1a.

We thank the reviewer for its suggestion. We substituted L1a with L1 in the text. Particularly, we substituted L1 to L1a in new line 30 and line 73.

9. It would be helpful to highlight the uniqueness of this work by explaining the limit of existing WNV data available, and how the new WNV genomes sequenced by this study could uniquely provide the insights/opportunity to assess such hypotheses.

We thank the reviewer for these important suggestions that we believe greatly improved our work. We added the uniqueness of our study to new lines 394-410, as follows:

“Mapping lineage dispersal can be an important tool to help identify potential sources of infections, track the direction and speed of spread, and reveal areas of high transmission or disease hotspots, offering valuable information for epidemiological investigations and shedding light on the ecological and environmental processes. Our work constitutes an important step in this direction. It gives an integration of extensive and diverse data obtained both from Italy and from West Africa (Senegal) and reinforces prior studies and findings highlighting the genetic relationships and circulation dynamics of the two strains between European and African countries. In particular, we collected strains that are genetically distinct from the ones publicly available (e.g. ON813216|Senegal|2012/08/31* and

ON813215|Senegal|2012/08/31* for WNV L1 and OQ204315|Italy|2022/11/02*), which allow us to improve the final phylogenetic and phylogeographic inference. By analysing the evolutionary relationships of the virus through phylogenetics and incorporating spatial information through phylogeography, we gain valuable insights into how the two viral lineages have independently evolved and dispersed across space and time. This knowledge has practical implications for surveillance and preparedness efforts concerning future outbreaks of WNV and other viruses, enhancing our ability to monitor and respond to WNV outbreaks effectively, and ultimately contributing to public health and disease control strategies”.

Reviewer #3:

“This is a very well and clearly written manuscript. I am not a specialist in Genomic analysis, so cannot really comment effectively on the methodologies or any specific aspects of it, but as this technology is fairly well established by now, I doubt there are any major methodological comments needed. My one substantive comment concerns the vectors - in the discussion you speculate that the difference in spread pattern/rate between L1 and L2 may be to do with the pathogens impact on bird hosts. I wonder whether the vector competence for the two strains varies significantly, and whether this might also have a differential impact on each lineage. My recommendation is to publish with minor revisions.”

We thank the reviewer for raising this very interesting point. Following your important suggestion, our discussion section has been updated (lines 371 – 393 and lines 436-439), as follows:

“Each bird species appears to be characterised by a variability in migration patterns, creating a complex network of connectivity between Africa and Europe (<https://migrationatlas.org/>), which may underlie the enormous and variable spread of WNV L1 and L2 strains worldwide. However, it is still difficult to understand the reason why WNV L1 spreads more efficiently than L2, given that both lineages have the same chance of infecting bird species. One possible explanation could be that WNV L1 and L2 infection may have different outcomes in these species^{5,38}. For instance, WNV L2 infection could result in a severe and fatal disease or in a very mild infection with brief and low levels of viraemia^{39,42}. In either case, the host would decrease its capability to transmit and spread the infection.

When investigating the diverse pattern of circulation of the two viral lineages, it is important to consider the involvement of vectors in viral dispersal. In fact, vectors play a crucial role in transmitting viruses during their blood meal⁴³. Most human outbreaks worldwide have been caused by L1 strains despite the strong presence of L2 in Europe and some African countries^{10,15,31,32,43}, and the existence of other local strains, such as L8 and KOUTV in Africa^{44,45}, or the Rabensburg virus and L4 in Europe^{46,47}. While WNV L1 and L2 seem to have a minor impact in Africa¹⁰, the introduction of these African strains in Europe have caused a significant number of severe WNV cases in humans and other animals^{31,32,43}, showing that local strains could have the ability to adapt to new environments and ecological niches and therefore be transmitted and become established. Based on the geographic area, vector species, such as mosquitoes, vary. Depending on factors such as genetic variations, environmental conditions, and evolutionary pressures, different geographic populations are characterised by a diverse behavior and susceptibility to arbovirus infection⁴⁸. In Africa, Gamou Fall et al. compared the L1 and L2 vector competence of African *Cx. neavei* and *Cx. quinquefasciatus* mosquitoes, often found naturally infected by WNV in Senegal, showing better transmission of L1 compared to L2⁴⁵. In Europe, competent studies performed on *Cx. pipiens* mosquitoes, the major WNV competent vectors in the continent, showed these mosquitoes to be able transmitters of both L1 and L2 strains⁴⁸. It clearly appears that the differential circulation of these two strains in Europe and Africa is correlated to multiple factors, as the ability of a mosquito to transmit the virus, the mosquito population density, the host feeding preferences, the availability of susceptible amplifying hosts, the virus divergence coming from the interaction between the virus and the avian host's immune responses; the diverse pathogenicity of circulating strains; and the association with human and animal populations⁴⁸.

Lines “424-427”:

“Overall, the enhancement of an integrated and homogeneous surveillance plan in Europe and Africa would yield valuable benefits by allowing the generation of new consistent datasets including newly generated genomes and sequence metadata from previously unreported countries and regions. To unravel the L1 and L2 strain dynamics, **it would be essential to combine these new datasets with studies on vector competence of diverse WNV strains, alongside with investigations into the abundance and host feeding preferences of various mosquito species.** Finally, incorporating GPS data from migratory birds and studies on different bird species susceptibility would further contribute to the understanding on how the

virus spread and which reservoirs play a significant role in its transmission across different regions, helping to mitigate the impact and predict future outbreaks, not only of this pathogen, but also of new emerging viruses.”

We carefully revised our manuscript, moving the “Material and Methods” section after the “Discussion” section. Also, we deposited all new data associated with the paper as Supplementary files in a publicly accessible repository (Figshare, project no. 160822) where they can be freely and enduringly accessed. Particularly, we included in this repository all new accession codes and metadata of West Nile virus genome sequences obtained by IZS-Teramo and IPD-Dakar deposited on NCBI and used in the study, and sequence alignment and tree files used for phylogenetic and phylogeographic analyses. This information has been added to the “Data Availability” section after the “Material and Methods” section but before the References, and can be found at:

Supplementary Tables DOI: at dx.doi.org/10.6084/m9.figshare.23660139

Supplementary Figures DOI: dx.doi.org/10.6084/m9.figshare.23660163

Alignments used for the phylogenetic analysis, model selection analysis, and tree files:
dx.doi.org/10.6084/m9.figshare.22182418.

Phylogeographic analyses DOI: dx.doi.org/10.6084/m9.figshare.23660571

The bibliography has been updated, as follows (Now lines 599-778):

1. Georgopoulou, I. & Tsiouris, V. The potential role of migratory birds in the transmission of zoonoses. *Vet Ital* 44, 671–677 (2008).
2. Mancuso, E. et al. West Nile and Usutu Virus Introduction via Migratory Birds: A Retrospective Analysis in Italy. *Viruses* 14, 416 (2022).
3. Komar, N. West Nile virus: epidemiology and ecology in North America. *Adv Virus Res* 61, 185–234 (2003).
4. Vidaña, B. et al. The Role of Birds of Prey in West Nile Virus Epidemiology. *Vaccines (Basel)* 8, (2020).

5. Pérez-Ramírez, E., Llorente, F. & Jiménez-Clavero, M. Á. Experimental infections of wild birds with West Nile virus. *Viruses* 6, 752–781 (2014).
6. Schwartz, G. et al. West Nile Virus in Common Wild Avian Species in Israel. *Pathogens* 11, 107 (2022).
7. Uelmen, J. A. et al. Human biting mosquitoes and implications for West Nile virus transmission. *Parasit Vectors* 16, 2 (2023).
8. García-Carrasco, J.-M., Muñoz, A.-R., Olivero, J., Segura, M. & Real, R. An African West Nile virus risk map for travellers and clinicians. *Travel Med Infect Dis* 52, 102529 (2022).
9. Gossner, C. M. et al. West Nile virus surveillance in Europe: moving towards an integrated animal-human-vector approach. *Eurosurveillance* 22, 30526 (2017).
10. Mencattelli, G. et al. Epidemiology of West Nile virus in Africa: An underestimated threat. *PLOS Neglected Tropical Diseases* 16, e0010075 (2022).
11. Dellicour, S. et al. Epidemiological hypothesis testing using a phylogeographic and phylodynamic framework. *Nat Commun* 11, 5620 (2020).
12. Baele, G., Dellicour, S., Suchard, M. A., Lemey, P. & Vrancken, B. Recent advances in computational phylodynamics. *Curr Opin Virol* 31, 24–32 (2018).
13. Lustig, Y. et al. Mosquito Surveillance for 15 Years Reveals High Genetic Diversity Among West Nile Viruses in Israel. *J Infect Dis* 213, 1107–1114 (2016).
14. Bakonyi, T. et al. Lineage 1 and 2 Strains of Encephalitic West Nile Virus, Central Europe. *Emerg Infect Dis* 12, 618–623 (2006).
15. Ndione, M. H. D. et al. Re-Introduction of West Nile Virus Lineage 1 in Senegal from Europe and Subsequent Circulation in Human and Mosquito Populations between 2012 and 2021. *Viruses* 14, 2720 (2022).
16. García-Carrasco, J.-M., Muñoz, A.-R., Olivero, J., Segura, M. & Real, R. Predicting the spatio-temporal spread of West Nile virus in Europe. *PLOS Neglected Tropical Diseases* 15, e0009022 (2021).
17. Barzon, L. et al. Rapid spread of a new West Nile virus lineage 1 associated with increased risk of neuroinvasive disease during a large outbreak in northern Italy, 2022: One Health analysis. *J Travel Med taac125* (2022) doi:10.1093/jtm/taac125.
18. Charrel, R. N. et al. Evolutionary relationship between Old World West Nile virus strains. Evidence for viral gene flow between Africa, the Middle East, and Europe. *Virology* 315, 381–388 (2003).

19. Hayes, E. B. et al. Epidemiology and Transmission Dynamics of West Nile Virus Disease. *Emerg Infect Dis* 11, 1167–1173 (2005).
20. Sotelo, E. et al. Phylogenetic relationships of Western Mediterranean West Nile virus strains (1996–2010) using full-length genome sequences: single or multiple introductions? *Journal of General Virology* 92, 2512–2522 (2011).
21. McMullen, A. R. et al. Molecular evolution of lineage 2 West Nile virus. *J Gen Virol* 94, 318–325 (2013).
22. Shah-Hosseini, N., Chinikar, S., Ataei, B., Fooks, A. R. & Groschup, M. H. Phylogenetic analysis of West Nile virus genome, Iran. *Emerging infectious diseases* 20, 1419 (2014).
23. Aguilera-Sepúlveda, P., Gómez-Martín, B., Agüero, M., Jiménez-Clavero, M. Á. & Fernández-Pinero, J. A new cluster of West Nile virus lineage 1 isolated from a northern goshawk in Spain. *Transbound Emerg Dis* 69, 3121–3127 (2022).
24. May, F. J., Davis, C. T., Tesh, R. B. & Barrett, A. D. T. Phylogeography of West Nile virus: from the cradle of evolution in Africa to Eurasia, Australia, and the Americas. *J. Virol.* 85, 2964–2974 (2011).
25. Santos, P. D. et al. An advanced sequence clustering and designation workflow reveals the enzootic maintenance of a dominant West Nile virus subclade in Germany. *bioRxiv* 2022–10 (2022).
26. Schuffenecker, I. et al. West Nile Virus in Morocco, 2003. *Emerg Infect Dis* 11, 306–309 (2005).
27. Mann, B. R., McMullen, A. R., Swetnam, D. M. & Barrett, A. D. T. Molecular Epidemiology and Evolution of West Nile Virus in North America. *Int J Environ Res Public Health* 10, 5111–5129 (2013).
28. Hernández-Triana, L. M. et al. Emergence of West Nile Virus Lineage 2 in Europe: A Review on the Introduction and Spread of a Mosquito-Borne Disease. *Front. Public Health* 2, (2014).
29. Tantely, M. L., Goodman, S. M., Rakotondranaivo, T. & Boyer, S. Review of West Nile virus circulation and outbreak risk in Madagascar: Entomological and ornithological perspectives. *Parasite* 23, 49 (2016).
30. Srihi, H., Chatti, N., Ben Mhadheb, M., Gharbi, J. & Abid, N. Phylodynamic and phylogeographic analysis of the complete genome of the West Nile virus lineage 2 (WNV-2) in the Mediterranean basin. *BMC Ecology and Evolution* 21, 183 (2021).
31. Beck, C. et al. Contrasted Epidemiological Patterns of West Nile Virus Lineages 1 and 2 Infections in France from 2015 to 2019. *Pathogens* 9, (2020).

32. Mencattelli, G. et al. Epidemiological and Evolutionary Analysis of West Nile Virus Lineage 2 in Italy. *Viruses* 15, 35 (2023).
33. Zehender, G. et al. Reconstructing the recent West Nile virus lineage 2 epidemic in Europe and Italy using discrete and continuous phylogeography. *PLoS ONE* 12, e0179679 (2017).
34. Broad-scale patterns of the Afro-Palaeartic landbird migration - Briedis - 2020 - Global Ecology and Biogeography - Wiley Online Library. <https://onlinelibrary.wiley.com/doi/abs/10.1111/geb.13063>.
35. Rappole, J. Migratory Birds and Spread of West Nile Virus in the Western Hemisphere. *Emerg. Infect. Dis.* 6, 319–328 (2000).
36. Guilherme, J. L. et al. Connectivity between countries established by landbirds and raptors migrating along the African-Eurasian flyway. *Conserv Biol* 37, e14002 (2023).
37. An Integrative Eco-Epidemiological Analysis of West Nile Virus Transmission - PubMed. <https://pubmed.ncbi.nlm.nih.gov/28584951/>.
38. Tolsá, M. J., García-Peña, G. E., Rico-Chávez, O., Roche, B. & Suzán, G. Macroecology of birds potentially susceptible to West Nile virus. *Proc Biol Sci* 285, 20182178 (2018).
39. Vaughan, J. A., Newman, R. A. & Turell, M. J. Bird species define the relationship between West Nile viremia and infectiousness to *Culex pipiens* mosquitoes. *PLOS Neglected Tropical Diseases* 16, e0010835 (2022).
40. Gamino, V. & Höfle, U. Pathology and tissue tropism of natural West Nile virus infection in birds: a review. *Vet Res* 44, 39 (2013).
41. Pathogenesis of West Nile Virus Lineage 2 in Domestic Geese after Experimental Infection - PubMed. <https://pubmed.ncbi.nlm.nih.gov/35746790/>.
42. Mencattelli, G. et al. West Nile Virus Lineage 2 Overwintering in Italy. *Tropical Medicine and Infectious Disease* 7, 160 (2022).
43. Bakonyi, T. et al. Explosive spread of a neuroinvasive lineage 2 West Nile virus in Central Europe, 2008/2009. *Veterinary Microbiology* 165, 61–70 (2013).
44. Fall, G. et al. First Detection of the West Nile Virus Koutango Lineage in Sandflies in Niger. *Pathogens* 10, 257 (2021).
45. Fall, G., Diallo, M., Loucoubar, C., Faye, O. & Sall, A. A. Vector competence of *Culex neavei* and *Culex quinquefasciatus* (Diptera: Culicidae) from Senegal for lineages 1, 2, Koutango and a putative new lineage of West Nile virus. *Am. J. Trop. Med. Hyg.* 90, 747–754 (2014).

46. Lvov, D. K. et al. West Nile virus and other zoonotic viruses in Russia: examples of emerging-reemerging situations. *Arch Virol Suppl* 85–96 (2004) doi:10.1007/978-3-7091-0572-6_7.
47. Hubálek, Z., Halouzka, J., Juricová, Z. & Sebesta, O. First isolation of mosquito-borne West Nile virus in the Czech Republic. *Acta Virol* 42, 119–120 (1998).
48. Fiacre, L. et al. Molecular Determinants of West Nile Virus Virulence and Pathogenesis in Vertebrate and Invertebrate Hosts. *Int J Mol Sci* 21, 9117 (2020).
49. Fros, J. J. et al. West Nile Virus: High Transmission Rate in North-Western European Mosquitoes Indicates Its Epidemic Potential and Warrants Increased Surveillance. *PLoS Negl Trop Dis* 9, e0003956 (2015).
50. Traoré-Lamizana, M. et al. Arbovirus surveillance from 1990 to 1995 in the Barkedji area (Ferlo) of Senegal, a possible natural focus of Rift Valley fever virus. *J. Med. Entomol.* 38, 480–492 (2001).
51. Barry, M. A. et al. Performance of case definitions and clinical predictors for influenza surveillance among patients followed in a rural cohort in Senegal. *BMC Infectious Diseases* 21, 1–11 (2021).
52. Digoutte, J. P., Calvo-Wilson, M. A., Mondo, M., Traore-Lamizana, M. & Adam, F. Continuous cell lines and immune ascitic fluid pools in arbovirus detection. *Res Virol* 143, 417–422 (1992).
53. Del Amo, J. et al. A novel quantitative multiplex real-time RT-PCR for the simultaneous detection and differentiation of West Nile virus lineages 1 and 2, and of Usutu virus. *Journal of Virological Methods* 189, 321–327 (2013).
54. Fall, G. et al. Real-Time RT-PCR Assays for Detection and Genotyping of West Nile Virus Lineages Circulating in Africa. *Vector Borne Zoonotic Dis.* 16, 781–789 (2016).
55. Mencattelli, G. et al. West Nile Virus Lineage 1 in Italy: Newly Introduced or a Re-Occurrence of a Previously Circulating Strain? *Viruses* 14, 64 (2022).
56. SARS-CoV-2 surveillance in Italy through phylogenomic inferences based on Hamming distances derived from pan-SNPs, -MNPs and -InDels - PubMed. <https://pubmed.ncbi.nlm.nih.gov/34717546/>.
57. Grubaugh, N. D. et al. An amplicon-based sequencing framework for accurately measuring intrahost virus diversity using PrimalSeq and iVar. *Genome Biology* 20, 8 (2019).
58. Capella-Gutiérrez, S., Silla-Martínez, J. M. & Gabaldón, T. trimAl: a tool for automated alignment trimming in large-scale phylogenetic analyses. *Bioinformatics* 25, 1972–1973 (2009).

59. Martin, D. P., Murrell, B., Golden, M., Khoosal, A. & Muhire, B. RDP4: Detection and analysis of recombination patterns in virus genomes. *Virus Evol* 1, vev003 (2015).
60. Martin, D. & Rybicki, E. RDP: detection of recombination amongst aligned sequences. *Bioinformatics* 16, 562–563 (2000).
61. Padidam, M., Sawyer, S. & Fauquet, C. M. Possible emergence of new geminiviruses by frequent recombination. *Virology* 265, 218–225 (1999).
62. Salminen, M. O., Carr, J. K., Burke, D. S. & McCutchan, F. E. Identification of breakpoints in intergenotypic recombinants of HIV type 1 by bootscanning. *AIDS Res Hum Retroviruses* 11, 1423–1425 (1995).
63. Smith, J. M. Analyzing the mosaic structure of genes. *J Mol Evol* 34, 126–129 (1992).
64. Posada, D. & Crandall, K. A. Evaluation of methods for detecting recombination from DNA sequences: Computer simulations. *Proceedings of the National Academy of Sciences* 98, 13757–13762 (2001).
65. Gibbs, M. J., Armstrong, J. S. & Gibbs, A. J. Sister-scanning: a Monte Carlo procedure for assessing signals in recombinant sequences. *Bioinformatics* 16, 573–582 (2000).
66. Boni, M. F., Posada, D. & Feldman, M. W. An exact nonparametric method for inferring mosaic structure in sequence triplets. *Genetics* 176, 1035–1047 (2007).
67. Kalyaanamoorthy, S., Minh, B. Q., Wong, T. K. F., von Haeseler, A. & Jermini, L. S. ModelFinder: fast model selection for accurate phylogenetic estimates. *Nat Methods* 14, 587–589 (2017).
68. Stamatakis, A. RAxML version 8: a tool for phylogenetic analysis and post-analysis of large phylogenies. *Bioinformatics* 30, 1312–1313 (2014).
69. Poon, A. F. Y. et al. Reconstructing the Dynamics of HIV Evolution within Hosts from Serial Deep Sequence Data. *PLoS Comput Biol* 8, e1002753 (2012).
70. Rambaut, A., Lam, T. T., Max Carvalho, L. & Pybus, O. G. Exploring the temporal structure of heterochronous sequences using TempEst (formerly Path-O-Gen). *Virus Evol* 2, vew007 (2016).
71. Kass, R. E. & Raftery, A. E. Bayes Factors. *Journal of the American Statistical Association* 90, 773–795 (1995).
72. Suchard, M. A. et al. Bayesian phylogenetic and phylodynamic data integration using BEAST 1.10. *Virus Evol* 4, vey016 (2018).
73. Rambaut, A., Drummond, A. J., Xie, D., Baele, G. & Suchard, M. A. Posterior Summarization in Bayesian Phylogenetics Using Tracer 1.7. *Syst Biol* 67, 901–904 (2018).

74. Spread3: Interactive Visualization of Spatiotemporal History and Trait Evolutionary Processes - PubMed. <https://pubmed.ncbi.nlm.nih.gov/27189542/>. Georgopoulou, I. & Tsiouris, V. The potential role of migratory birds in the transmission of zoonoses. *Vet Ital* 44, 671–677 (2008).

The supplementary materials have been updated, as follows (Now lines 812-885):

Supplementary Figure 1. Maximum likelihood phylogeny of 228 WNV L1 genomes. Supplementary Figure 1 shows the Maximum likelihood phylogeny of West Nile virus lineage 1 genomes used in this paper. Nodes with bootstrap supports (BS) lower than 90 are depicted with light orange dots. A scale bar at the bottom of the figure indicates the number of substitutions per site. All samples belong to clade 1, which is divided into different clusters (C1-C7) highlighted with colour bars on the right. Other relevant groups mentioned in the discussion are annotated in the same way: Eastern-European clade (EEC), Western Mediterranean clade (WMed, which is divided into 2 subclades, WMed1 and WMed2) and the Italian clade sampled near Livenza in 2011-2013.

Supplementary Figure 2. Maximum likelihood phylogeny of 297 WNV L2 genomes. Supplementary Figure 2 shows the Maximum likelihood phylogeny of West Nile virus lineage 2 genomes used in this paper. Nodes with bootstrap supports (BS) lower than 50 are depicted with light orange dots. A scale bar at the bottom of the figure indicates the number of substitutions per site. Clades (2a-d) are indicated by coloured bars on the right side of the figure. Clade 2d is further divided into 5 clusters, annotated with black bars on the right side of the figure.

Supplementary Figure 3. Downsampling of the WNV L1 dataset. Midpoint-rooted tree for the maximum likelihood analysis of 228 WNV L1 genomes is shown. Sequences that were selected for phylogeographic analysis are coloured in green. Nodes with bootstrap supports (BS) lower than 90 are depicted with light orange dots. Nodes with no bootstrap are indicated with a black dot.

Supplementary Figure 4. Downsampling of the WNV L2 dataset. Midpoint-rooted tree for the maximum likelihood analysis of 297 WNV L2 genomes is shown. Sequences that were selected for phylogeographic analysis are coloured in green. Nodes with bootstrap supports (BS) lower than 90 are depicted with light orange dots. Nodes with no bootstrap are indicated with a black dot.

Supplementary Figure 5. Root-to-tip divergence analysis WNV L1. A summary of the root-to-tip divergence analysis for the WNV L1 full dataset used for the maximum likelihood analysis (a, b, c) and for the reduced one used for Bayesian phylogeographic inference (d, e, f) is shown

in the figure. a) Root-to-tip regression line showing, showing a positive correlation between root-to-tip divergence and time. b) Residuals of the regression analysis. c) Positive relation between node density and time.

Supplementary Figure 6. Root-to-tip divergence analysis WNV L2. Root-to-tip divergence analysis WNV L1. A summary of the root-to-tip divergence analysis for the WNV L2 full dataset used for the maximum likelihood analysis (a, b, c) and for the reduced one used for Bayesian phylogeographic inference (d, e, f) is shown in the figure. a) Root-to-tip regression line showing, showing a positive correlation between root-to-tip divergence and time. b) Residuals of the regression analysis. c) Positive relation between node density and time.

Supplementary Figure 7. Scatterplot showing the coordinates (latitude and longitude expressed in decimal degrees) of the inferred ancestral node locations, for the full dataset and for its different subsets sampled in the sensitivity analysis (different retained subsets of the main dataset are indicated by colour and shape: 100%: black dot; 75%: green cross; 50%: yellow square; 20%: light-red triangle). Data for WNV L1 a) and WNV L2 tend to overlap, with small differences among all datasets.

Supplementary Figure 8. Sensitivity analysis for testing the robustness of the phylogeographic inference of WNV L1. Phylogeographic reconstructions for three subsets of the main dataset (a) 75%, b) 50% and c) 30%) returned very similar scenarios, always showing the presence of a corridor that virtually connects Senegal, Morocco, and southern Europe. Red shadows indicate the 80%HPDs of the inferred location, coloured by the inferred median date for that area, with darker hue indicating older areas of circulation.

Supplementary Figure 9. Sensitivity analysis for testing the robustness of the phylogeographic inference of WNV L2. Phylogeographic reconstructions for three subsets of the main dataset (a) 75%, b) 50% and c) 30%) returned very similar scenarios, with a major introductory event from southern Africa to central-eastern Europe (with the exception of b), in which the direction is reversed, as the random sampling step included only very recent African sequences, without old genomes). Red shadows indicate the 80%HPDs of the inferred location, coloured by the inferred median date for that area, with darker hue indicating older areas of circulation.

Supplementary Table 1. West Nile virus genome sequences curated metadata. A table of curated metadata of West Nile virus lineages 1 and 2 obtained at IZS-Teramo and IPD-Dakar and used in the paper.

Supplementary Table 2. West Nile virus genome sequences downloaded from NCBI. A table of curated metadata of West Nile virus lineages 1 and 2 genome sequences obtained from NCBI and used in the paper.

Supplementary Table 3. Geographic coordinates of West Nile virus sequences used for molecular clocks. A table of geographic coordinates of West Nile virus lineages 1 and 2 sequences used for molecular clock analyses.

Supplementary Table 4. Results of the RDP4 analysis, showing all output parameters for detected recombinants.

Supplementary Table 5. Table showing the full results of the root-to-tip regression analyses of all WNV L1 and WNV L2 datasets analysed in the paper.

Supplementary Table 6. Table showing the log Bayes factors obtained from subtracting to the -log values of the estimated marginal likelihood of the most favourite model (relaxed clock with a Bayesian Skyline tree prior) the same estimated values for all the other models.

REVIEWERS' COMMENTS

Reviewer #2 (Remarks to the Author):

The authors have addressed all my comments. There are substantial improvement in the figures and tables.

I have the following minor points to further improve the manuscript:

1. Line 587-588: "... MCMC length (250*10⁶ generations sampling every 2.5*10⁴ steps and 200*10⁶ generations sampling every 2*10⁴ steps for the 30% subsample)." The 10⁴ and 10⁶ are actually referring to ten to power 4 and power 6, i.e. 10⁴, 10⁶, right? Please don't forget the superscript.

2. I would prefer not to say "molecular clocking", but "molecular clock analysis" which is more formal.

3. Figure 1 and 2. It is indicated that the yellow dots refer to the node with <0.9 posterior clade probability. Is it a typo that the correct meaning should be >0.9? It is a more usual practice for using nodes to indicate strongly supported node such as posterior clade probability > 0.9. Please clarify and correct if necessary.

4. The statements added in the revision "We overcame this limitation by converting into corresponding geographic coordinates the descriptive sampling location, such as district, province, or region.". It reads strangely, especially the "geographical coordinates the descriptive sampling location". It seems there are words missing from that, and the plural/singular form is used incorrectly. In fact, I would highly recommend the authors and journal editors to revise the whole manuscript for better use of language.

Tommy Lam

Response to reviewers

Reviewer #2 (Remarks to the Author):

The authors have addressed all my comments. There are substantial improvement in the figures and tables.

Thank you very much for the suggestions and for the time invested in reviewing this work. We think all comments were spot on and accurate, making it possible to improve the quality of the paper.

I have the following minor points to further improve the manuscript:

1. Line 587-588: "... MCMC length (250*10⁶ generations sampling every 2.5*10⁴ steps and 200*10⁶ generations sampling every 2*10⁴ steps for the 30% subsample)." The 10⁴ and 10⁶ are actually referring to ten to power 4 and power 6, i.e. 10⁴, 10⁶, right? Please don't forget the superscript.

Exactly, thank you for pointing out that the superscript was lacking. We corrected all numbers with the right notation (Now lines 557 – 558).

2. I would prefer not to say "molecular clocking", but "molecular clock analysis" which is more formal.

We replaced “molecular clocking” with “molecular clock analysis” (New line 556).

3. Figure 1 and 2. It is indicated that the yellow dots refer to the node with <0.9 posterior clade probability. Is it a typo that the correct meaning should be >0.9? It is a more usual practice for using nodes to indicate strongly supported node such as posterior clade probability > 0.9. Please clarify and correct if necessary.

The value in the main figures is not a typo, so all the values shown are correct. In this case, we decided to show only the low supported nodes for two reasons:

- 1) Most of the nodes are very well supported (posterior ≥ 0.9), so it is much easier to show the few problematic ones
- 2) We also tried the opposite visualisation (highlighting all nodes with posterior ≥ 0.9), but this resulted in a tree which is less easy to read. In particular, time bars are covered by the dots and it becomes harder to interpret the figure.

4. The statements added in the revision "We overcame this limitation by converting into corresponding geographic coordinates the descriptive sampling location, such as district, province, or region." It reads strangely, especially the "geographical coordinates the descriptive sampling location". It seems there are words missing from that, and the plural/singular form is used incorrectly. In fact, I would highly recommend the authors and journal editors to revise the whole manuscript for better use of language.

Tommy Lam

Thank you for the feedback on language. We rephrased the sentence highlighted in the comment as follows: “Having accurate geographic coordinates and collection dates for isolates collected over a significant period of time is essential for precisely calibrating molecular clock and phylogeographic models. This accuracy allows for dependable estimation of when and where epidemic events occurred. To address this challenge, we transformed descriptive sampling locations (like districts, provinces, or regions) into their corresponding geographic coordinates. Additionally, we addressed cases where collection dates were missing by attempting to retrieve them from the relevant research papers, if available. In situations where the collection date couldn't be identified, we assigned an estimated date. If at least the year was known, we selected the average date calculated from samples with complete information.” (New Lines 389 – 398).

We further checked the paper for the use of language.